# Trifloxystrobin blocks the growth of Theileria parasites and is a promising drug to treat Buparvaquone resistance

Marie Villares[1,4], Nelly Lourenço[1,4], Jeremy Berthelet[1,4], Suzanne Lamotte[2], Leslie Regad[3], Souhila Medjkane[1], Eric Prina[2], Fernando Rodrigues-Lima[3], Gerald F. Späth [2] & Jonathan B. Weitzman[1✉]

*Theileria* parasites are responsible for devastating cattle diseases, causing major economic losses across Africa and Asia. *Theileria* spp. stand apart from other apicomplexa parasites by their ability to transform host leukocytes into immortalized, hyperproliferating, invasive cells that rapidly kill infected animals. The emergence of resistance to the theilericidal drug Buparvaquone raises the need for new anti-*Theileria* drugs. We developed a microscopy-based screen to reposition drugs from the open-access Medicines for Malaria Venture (MMV) Pathogen Box. We show that Trifloxystrobin (MMV688754) selectively kills lymphocytes or macrophages infected with *Theileria annulata* or *Theileria parva* parasites. Trifloxystrobin treatment reduced parasite load in vitro as effectively as Buparvaquone, with similar effects on host gene expression, cell proliferation and cell cycle. Trifloxystrobin also inhibited parasite differentiation to merozoites (merogony). Trifloxystrobin inhibition of parasite survival is independent of the parasite TaPin1 prolyl isomerase pathway. Furthermore, modeling studies predicted that Trifloxystrobin and Buparvaquone could interact distinctly with parasite Cytochrome B and we show that Trifloxystrobin was still effective against Buparvaquone-resistant cells harboring *TaCytB* mutations. Our study suggests that Trifloxystrobin could provide an effective alternative to Buparvaquone treatment and represents a promising candidate for future drug development against *Theileria* spp.

[1] Université Paris Cité, Epigenetics and Cell Fate, CNRS, F-75013 Paris, France. [2] Institut Pasteur, Université Paris Cité, INSERM U1201, Molecular Parasitology and Signaling Unit, Paris, France. [3] Université Paris Cité, BFA, UMR 8251, CNRS, ERL U1133, Inserm, F-75013 Paris, France. [4]These authors contributed equally: Marie Villares, Nelly Lourenço, Jeremy Berthelet. ✉email: jonathan.weitzman@u-paris.fr

*Theileria* spp. belong to the phylum of Apicomplexa parasites that cause important zoonotic and human diseases, including malaria (*Plasmodium* spp.) and toxoplasmosis (*Toxoplasma gondii*)[1–3]. The global burden of mortality from apicomplexa infections has driven international efforts to create collaborative drug-discovery programs. The open-access Medicines for Malaria Venture (MMV) Pathogen Box includes four hundred compounds that have been tested against a wide range of parasites and pathogens[4]. Here, we describe the re-purposing of compounds from the anti-malarial initiative to screen for drugs against *Theileria* spp.

The intracellular parasite *Theileria* spp. causes tick-borne diseases in sub-Saharan Africa, Asia and the Middle East with significant socio-economic impact on livestock; Tropical Theileriosis, also present in southern Europe and North Africa, is caused by *T. annulata* infections and East Coast Fever is caused by *T. parva*[2,3]. Current treatment regimens include the theilericidal drug Buparvaquone (also known as BW720C)[5], but reports of the emergence of drug-resistant strains in Tunisia, Iran, Turkey and Sudan raise new concerns for disease management[6–8]. Point mutations in *Theileria* cytochrome b were described in some cases[9,10], while we recently discovered mutations in the parasite gene encoding TaPin1 in Buparvaquone-resistant isolates from Tunisia and Sudan[6,7]. Thus, there is a need to identify drugs that are effective against both *T. parva* and *T. annulata* species with a new mode of action that does not involve these targets prone to resistance acquisition.

The two transforming *Theileria* species (*T. parva* and *T. annulata*) have the remarkable ability to induce cancer phenotypes in infected myeloid cells and lymphocytes and several studies have described mechanisms by which the parasites manipulate their host cells[1,11,12]. These include the induction of signaling pathways (such as JNK/AP-1 or Myc), the sequestration of host proteins (such as IKK or EB1) and the secretion of effectors[11,13–19]. The TaPin1 prolyl isomerase is secreted by intracellular *Theileria* parasites and contributes to their transforming ability by activating key signaling pathways:[20] the TaPin1-Fbw7 pathway leads to oncogenic proliferation and the TaPin1-PKM2 pathway activates glycolytic enzymes important for metabolic reprogramming[6,21]. There are likely other mechanisms that contribute to the hijacking of host cell functions[12]. Identifying new anti-*Theileria* compounds will offer new therapeutic options and will provide molecular tools for a chemical genetics approach to investigate host-parasite interactions.

Here, we describe the development of a microscopy-based screening strategy to identify compounds that could be effective drugs against *Theileria*. We screened compounds from the MMV Pathogen Box library and found several that killed B cells infected with *T. annulata* parasites. Subsequent characterization, including parasite lethality, host cell proliferation and gene expression, narrowed down the most promising compounds to Trifloxystrobin (MMV688754), a broad-spectrum fungicide widely used to treat plant pathogens. Importantly, we also demonstrate that MMV688754 acts independently of the TaPin1 mechanism and is effective against Buparvaquone-resistant cells with mutations in the parasite *Cytochrome b* (*CytB*) gene. These findings open new possibilities for the treatment of drug-resistant strains.

## Results

### A microscopy-based screen for MMV Pathogen Box compounds against *Theileria*-infected macrophages.

We developed a microscopy-based strategy for screening the MMV Pathogen Box library of compounds that allowed for high-content assessment of the drug impact. We chose to use the *T. annulata*-infected bovine macrophage cell line TaC12, because of ease of imaging and parasite monitoring. We used a derived cell line expressing a GFP-CLASP1 fusion protein, a host microtubule-stabilizing protein that is recruited to the schizont surface and allows visualization of intracellular parasites[22]. Finally, we also included staining with an antibody against histone H3K18 methylation (H3K18me1), an epigenetic mark that is exclusive to parasite nuclei[23]. In this way we could automate the screening for MMV Pathogen Box compounds (as previously developed for anti-*Leishmania* drugs[24]) and generate screening information for host cell numbers (DAPI imaging of host nuclei), schizont shape and distribution (Green, GFP imaging) and parasite nuclei numbers per host cell (Red, anti-H3K18me1 imaging) [Fig. 1a]. We expect that compounds which kill the parasite would also result in host cell apoptosis and reduced host cell survival[25]. We conducted an initial screen at 10 μM compound concentrations, incubated for 48 h, and assessed host cell survival by comparing host cell nuclei numbers with DMSO-treated control samples [Fig. 1b]. This experiment identified 44 compounds that gave survival rates for host cells of <25%. We then repeated the screen at 2 μM concentrations [Fig. 1b] and identified 9 compounds that killed host cells even at the lower concentration, highlighting the sensitivity and specificity of the screen. We compared these hit compounds with those from previously reported screens of the MMV library against *T. annulata*, *T. parva* or *T. equi* parasites [Fig. 1c][4,26–28]. Interestingly, these previous screens exhibited very little overlap in the hit compounds identified, but at least four of the compounds we identified were also found in previous studies (MMV003152, MMV675968, MMV676600 and MMV688372). Buparvaquone (MMV680480) was not picked up in our initial screen as it did not kill host cells to <25% survival; neither was it identified in the majority of the other MMV screening studies. Thus, our microscopy-based screen identified 9 promising compounds that kill infected host cells at 2 μM, some of which are consistent with previous screening results using alternative methodologies.

### MMV668754/Trifloxystrobin effectively reduced parasite and host cell survival of *Theileria*-infected lymphocytes.

One advantage of our screening strategy is the additional information gleaned from the microscopic data. We assessed the impact of the 9 primary hit compounds (identified at 2 μM) on parasite load per infected macrophage. Although all 9 compounds resulted in <25% host cell survival, only two compounds significantly reduced the number of parasites per host cell, as assessed by the number of H3K18me1-positive nuclei [Fig. 2a]. The most effective of these, by far, was the MMV668754 compound which corresponds to the drug Trifloxystrobin. This compound was more effective at reducing host cell survival and parasite cell number than Buparvaquone, the standard-of-care drug to treat *Theileria*-infected cows [Fig. 2b]. Some of the other 7 compounds caused a slight increase in the number of parasites per cell (which may be part of a general stress response). We chose to study the MMV668754/Trifloxystrobin compound in further detail.

To explore drug action further, we switched to additional *Theileria*-infected cell models. We tested the TBL3 vs BL3 paired bovine B sarcoma cell lines, that offers the advantage of comparing drug activity on infected vs uninfected cells. We treated *T. annulata*-infected TBL3 cells with MMV compounds from our primary screen and compared with uninfected BL3 controls. Testing the 9 MMV compounds from the initial macrophage TaC12 screen at 2 μM, we observed that 4 of the compounds reduced cell survival of infected TBL3 by >50% [Fig. 3a], but only two compounds (MMV688372 and MMV668754) demonstrated selective killing of infected TBL3 cells compared to uninfected cells (>80% survival) [Fig. 3a], with $EC_{50}$ values of 2.5 μM and 1.5 μM, respectively [Fig. 3b]. We examined the effects of these two compounds on the cell cycle of TBL3 and BL3 cells and compared

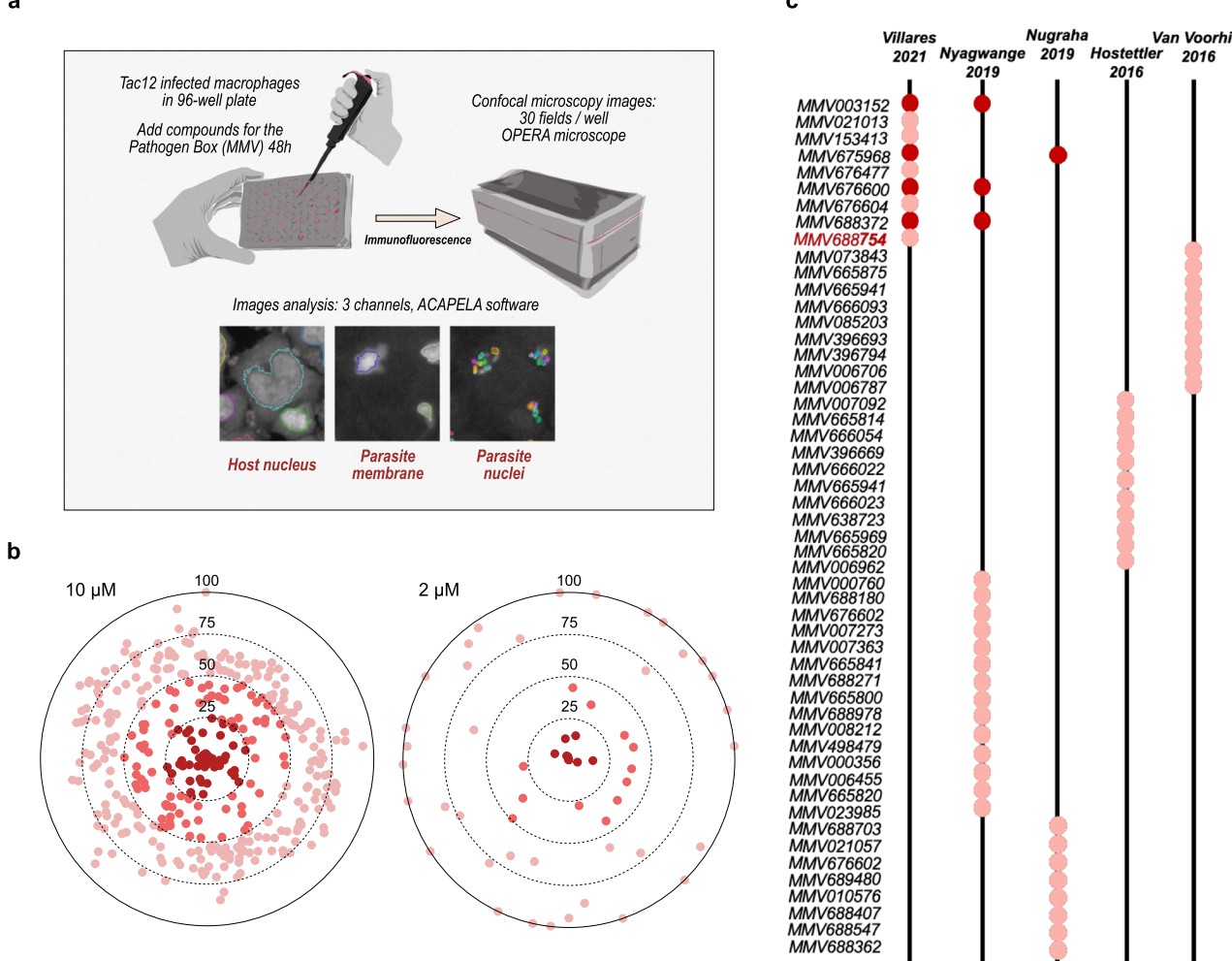

**Fig. 1 Screening MMV Pathogen Box compounds against *Theileria*-infected macrophages. a** A schematic presentation of the different steps of the microscopy-based drug screen. *T. annulata*-infected macrophages (TaC12 cell line) were distributed into 96-well plates and imaged after incubation (48 h) with compounds from the MMV Pathogen Box library at two concentrations. The screen involved identifying and segmentation of host cell nuclei (DAPI staining), the macroschizont membrane (CLASP-GFP fusion protein) and parasite nuclei (H3K18me1 staining). **b** TaC12 cells treated with MMV Pathogen Box compounds at two concentrations (10 μM or 2 μM) for 48 h. Host cell survival was calculated by comparing the number of host cell nuclei per field with control (DMSO-treated) cells. Results are plotted for percent host cell survival, with the darkest spots representing compounds causing <25% survival. **c** Comparison of our results with previous MMV screening reports for *Theileria*-infected cells. The hit compounds from screens conducted against cells infected with *T. equi* (Van Voorhis et al.[4], Nugraha et al.[28]), *T. parva* (Nyagwange et al.[26]) or *T. annulata* (Hostettler et al.[27]) are indicated as circles. Hits detected in more than one screen are indicated in dark red. The complete list of compounds is shown in Supplementary Table 1.

with Buparvaquone treatment. Buparvaquone was previously shown to induce a G1 growth-arrest in TBL3 lymphocytes, reduced S1 phase and increased apoptosis (assessed by sub-G1 population), with relatively little effect on uninfected BL3 cells[25,29] [Fig. 3c]. We observed similar effects for the MMV668754 compound (significant increases in G1 and sub-G1 populations) for TBL3 cells, but not BL3 cells [Fig. 3c]. This was not the case for the MMV688372 compound. Thus, the MMV668754/Trifloxystrobin drug appears to be as effective as Buparvaquone in its specific effects on parasite-infected lymphocytes.

We conducted further experiments comparing the effects of the MMV688372 and MMV668754 compounds with Buparvaquone treatment on *T. annulata*-infected macrophages (TaC12) and B lymphocytes (TBL3) and *T. parva*-infected lymphocytes (TpM) [Fig. 4]. We measured the average number of parasites per host cell in these three cellular scenarios and obtained similar results despite the different parasite species and different host cell types. MMV668754/Trifloxystrobin was as effective as Buparvaquone in significantly reducing parasite load in macrophages and

lymphocytes infected with *T. annulata* or *T. parva* parasites [Fig. 4a]. In contrast, the MMV688372 compound was similar to controls and did not affect parasite survival. Hemphill and colleagues recently proposed an elegant quantitative RT-PCR assay for assessing drug activities against intracellular *Theileria* schizonts[30]. We applied this TaSP assay to test drug impact on parasite viability and observed effective reduction in the *TaSP/bACTIN* mRNA ratios by treatment with MMV668754/Trifloxystrobin or Buparvaquone, but not MMV688372 or DMSO controls [Fig. 4b].

These combined results show that MMV668754/Trifloxystrobin is effective in killing infected bovine leukocytes and is comparable with Buparvaquone treatment in targeting *T. annulata* and *T. parva* schizonts in bovine lymphocytes and macrophages.

### MMV668754/Trifloxystrobin inhibits parasite-induced transformed phenotypes and parasite differentiation. A remarkable feature of *Theileria* spp is the ability to induce transformation of

**a**

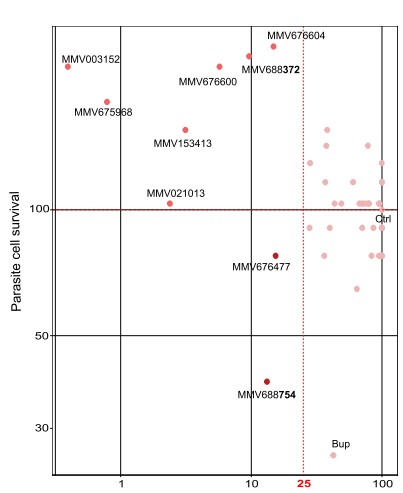

**b**

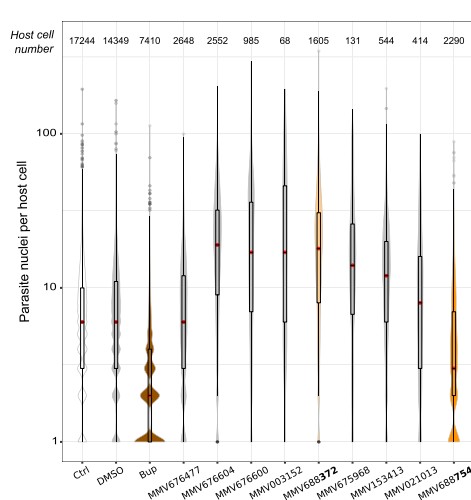

**Fig. 2 Screening for impact on parasite and host cell survival. a** Bi-parametric dot plots showing the host cell and parasite survival as a percentage of DMSO-treated controls. Only two MMV compounds decreased host and parasite cell survival effectively, while the other 7 caused increased parasite numbers. This is probably due to the stress caused by drug cytotoxicity of host cells and the impact on parasites in the remaining surviving cells. The DMSO control (Ctrl) and Buparvaquone (Bup) data are shown for comparison. **b** Violin plots of the number of parasites per host cell. Data are shown for untreated, DMSO-treated, buparvaquone (Bup)-treated and hit compound-treated samples. Only one drug (MMV688754) showed a strong decrease in parasite number comparable to Buparvaquone treatment. The relative surviving host cell numbers are indicated above. The red dot represents the median number of parasites per surviving host cell, the boxplot shows the 1st and 3rd quartiles. The Buparvaquone drug is shown in brown and the two compounds MMV688372 and MMV688754 used in subsequent experiments are colored orange.

host leukocytes[11,13]. Many studies have shown that Buparvaquone is an effective inhibitor of many of these transformed phenotypes, including proliferation and invasion[25,29,31,32]. In addition, several genes have been identified as key biomarkers of host cell transformation. We tested the effects of MMV668754 treatment on the expression of host genes in *T. annulata*-infected macrophages (TaC12) and B lymphocytes (TBL3). We found that MMV668754/Trifloxystrobin treatment was similar to Buparvaquone in its inhibition of host gene expression for the proliferative *oncomiR-155*, the invasive *MMP-9* gene and genes encoding metabolic enzymes Hexokinase 2 and PDK1 [Fig. 5a]. These genes are downstream of activated host transcriptional networks including c-Jun/AP-1 and HIF1α[25,29,31,33]. We also tested the ability of MMV668754/Trifloxystrobin to block host cell proliferation using a classic soft-agar colony growth assay. The colony growth of infected TBL3 cells was completely blocked by Buparvaquone and significantly reduced by adding MMV668754 [Fig. 5b]. In contrast, treatment with the MMV688372 drug had no effect on the expression of host genes induced by *Theileria*, or on colony growth of TBL3 cells [Fig. 5a–b]. Thus, MMV668754/Trifloxystrobin selectively reduced the gene expression and proliferation associated with parasite-induced transformation.

As MMV668754/Trifloxystrobin treatment did not cause extensive apoptosis in TaC12 macrophages we also tested the effect on the ability of surviving *Theileria* parasites to differentiate to merozoites [Fig. 6]. The merogony differentiation process can be induced by incubating infected macrophages at 41 °C for 8 days. Merogony was accompanied by a characteristic change in the size and number of parasite nuclei and induction of the merogony marker gene *TamR1*[34]. TaC12 cells were induced to merogony in the presence of different drugs. We observed a three-fold reduction in the number of macrophages containing expanded parasite numbers and we saw a significant reduction in *TamR1* expression in cells incubated with MMV668754/Trifloxystrobin [Fig. 6a], whereas the MMV688372 drug had no impact on parasite differentiation [Fig. 6b].

**MMV668754/Trifloxystrobin and Buparvaquone have distinct targets.** The experiments described above showed that MMV668754/Trifloxystrobin was equivalent to Buparvaquone in its ability to reduce parasite number and kill infected cells. We were interested in exploring the differences in mode of action of the two drugs. Buparvaquone was previously shown to target parasite TaPin1 prolyl isomerase and/or cytochrome B (CytB)[6,10,35]. The worrying emergence of Buparvaquone resistance was associated either with mutations in the *TaPin1* parasite gene that affected Buparvaquone inhibition of TaPIN1 catalytic activity[6,7] or with mutations in the parasite *CytB* gene[7,9,35]. Therefore, we tested whether MMV668754 also inhibited TaPin1 activity using a previously described in vitro assay[6]. We chose to compare MMV668754 with the DTM (dipentamethylene thiuram monosulfide) Pin1 inhibitor[6] as the latter is chemically distinct from Buparvaquone and MMV668754 [Fig. 7a]. We produced recombinant His-tagged TaPin1 protein and investigated drug specificity and isomerase activity [Fig. 7b]. DTM treatment effectively blocked TaPin1 activity in the in vitro assay, but MMV668754 had no effect [Fig. 7c]. We previously showed that the DTM inhibition of TaPin1 was unaffected by mutations that block Buparvaquone inhibition;[6] it seemed unnecessary to test these TaPin1 variants, as MMV668754 did not block wild-type activity. These results suggest that MMV668754/Trifloxystrobin inhibition of parasite survival is independent of the TaPin1 pathway.

The other known target of Buparvaquone is the parasite *CytB* gene that has been reported to be mutated in drug resistant strains that do no harbor *TaPin1* mutations[7,9,10]. Cytochrome B has also been described as a MMV668754/Trifloxystrobin target in other microorganisms[36]. We generated Buparvaquone-resistant cells by culturing infected TBL3 cells in the presence of Buparvaquone for an extended period of time. We sequenced the parasite *TaPin1* and *TaCytB* genes in the drug-resistant cells and identified three mutations in the *TaCytB* sequence [Fig. 8a]. Interestingly, at least one of these (V227M) was previously

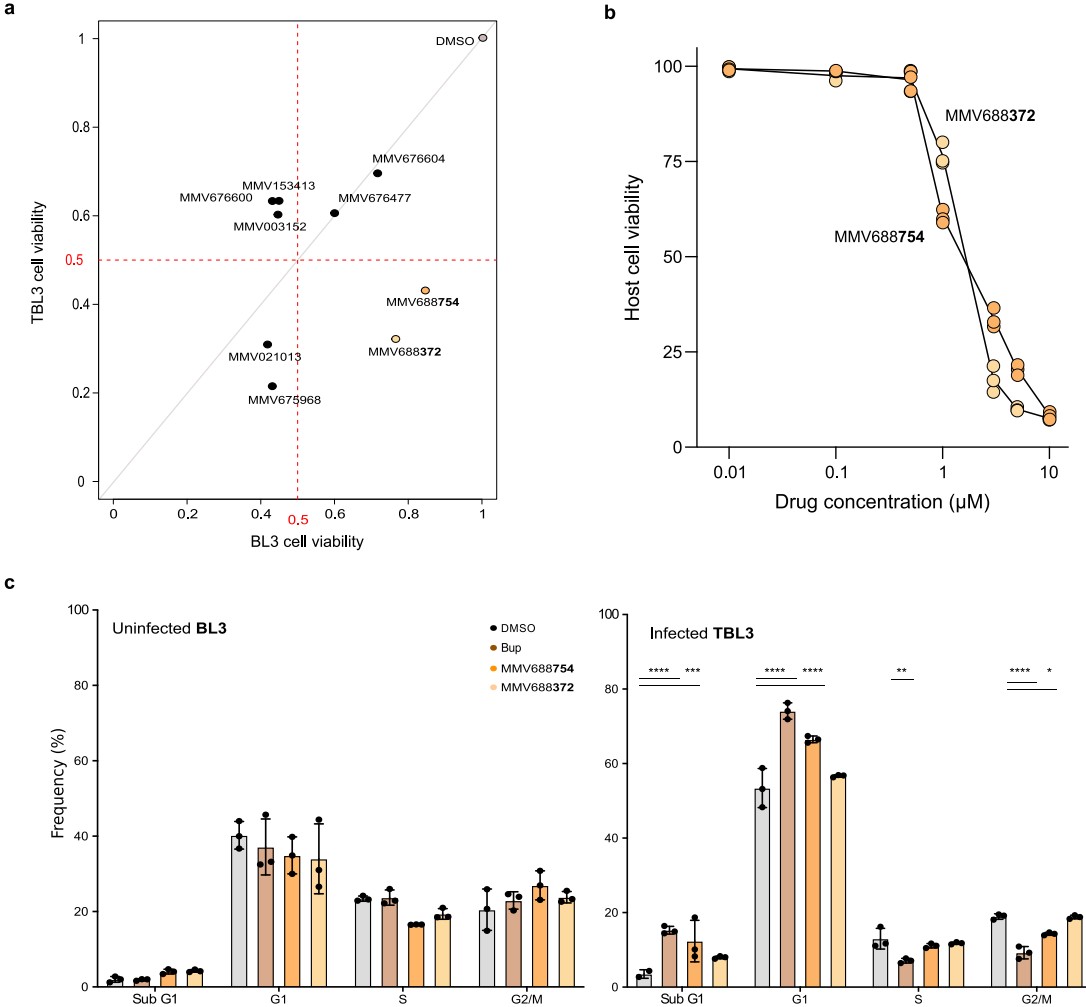

**Fig. 3 Comparing the drug impact on infected TBL3 vs uninfected BL3 cells. a** Bi-parametric dot plot showing the host cell viability in *Theileria*-TBL3 and uninfected BL3 cells with a 2 μM drug-treatment for 48 h. Only two MMV drugs reduced TBL3 survival (<50%), but not BL3 cells (around 80% survival). Representative of three independent experiments. **b** Cytotoxicity over a range of drug concentrations for TBL3 infected cells. The $IC_{50}$ for both drugs MMV688754 and MMV688372 were around 2 μM. Error bars represent mean values ± SD ($n = 3$ replicates). **c** Cell cycle analysis by flow cytometry performed on BL3 and TBL3 cells following treatment with 2 μM drugs for 48 h. Error bars represent mean values ± SD. $n = 3$ independent experiments. Two-way Anova statistical test, Dunnett's multiple comparison: *$p < 0.05$, **$p < 0.01$, ***$p < 0.001$, ****$p < 0.0001$.

reported in field samples of Buparvaquone resistance[9]. Our Buparvaquone-resistant cells did not show evidence of inactivating mutations in the *TaPin1* gene [Supplementary Fig. 1]. The Buparvaquone-resistant cells maintained a sensitivity to treatment with MMV668754/Trifloxystrobin and there was no additive effect of treatment with both drugs [Fig. 8b]. This suggests that the two drugs may have distinct modes of actions.

## Discussion

Anti-microbial drug resistance is emerging as a significant global problem in the treatment of disease. This is not restricted to human diseases, but extends to the large number of animal diseases that can wreak havoc on farms and livestock with considerable socio-economic impact. *Theileria* spp parasites cause major diseases in cattle and the emergence of parasite strains resistant to the widespread drug Buparvaquone highlights the need for alternative therapeutic options[7,9]. The compound libraries of the MMV initiative offer a trove to search for new anti-*Theileria* drugs with modes of action that are independent of Buparvaquone[4]. Here we describe a screen of the four hundred compounds of the MMV Pathogen Box and subsequent tests that

led to the identification of MMV668754/Trifloxystrobin as a novel anti-*Theileria* drug candidate.

Our study involved a series of increasingly stringent assays to select the more promising drug compound from the MMV Pathogen Box library[4]. We previously demonstrated that microscopy-based screens followed by careful phenotypic characterization present a powerful drug-discovery strategy[24]. Our microscopy screening approach recorded a number of parameters including host nuclei number, parasite schizont structure and the parasite load per host cell. By screening the Pathogen Box at 10 μM and 2 μM we initially focused on the more effective compounds (9 initially) with cytotoxicity against infected bovine macrophages. The subsequent criteria focused on parasite impact (decreased parasite survival per host cell) and infected vs uninfected selectivity (TBL3 vs BL3 cytotoxicity) led us to MMV668754 as the most promising candidate. The MMV668754 compound was as effective as the care-standard Buparvaquone drug in nearly all the assays we tested. Both compounds are effective against both *T. annulata* and *T. parva* species, as well as different host cell types (macrophages or lymphocytes). We also showed that MMV668754 treatment caused growth arrest and apoptosis and blocked colony growth in soft-agar assays. These

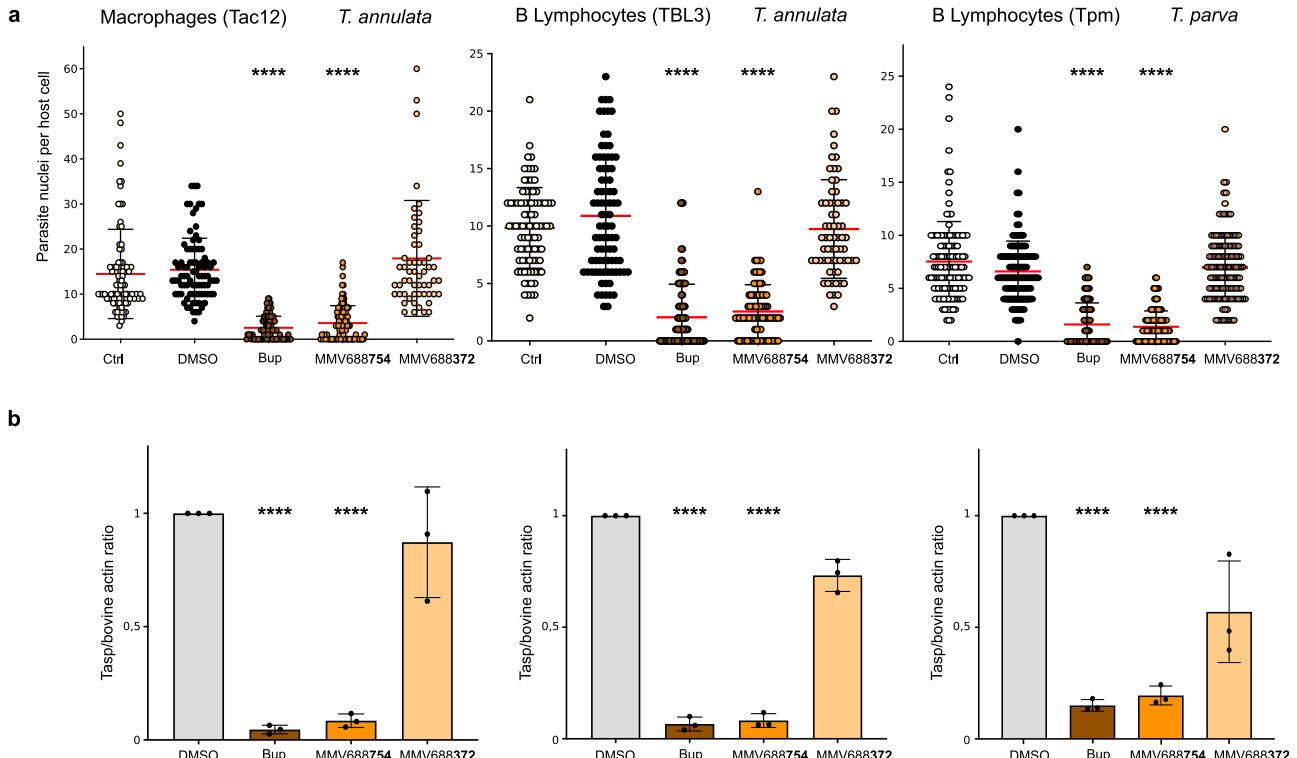

**Fig. 4 Drug effects on parasite number and gene expression in different parasite species and host cell types. a** Parasite load per infected host cell was evaluated after drug treatment (2 µM for 48 h) compared to untreated (Ctrl), Buparvaquone (Bup) or DMSO-treated cells. Host cells (at least $n = 50$) were assessed for each experiment and parasite nuclei (DAPI staining) per host cell were counted manually. These results are representative of 3 independent experiments. Kruskal Wallis non-parametric test followed by Dunn's multiple comparison test. ****$p < 0.0001$. Results are shown for *T. annulata*-infected macrophages (TaC12), B lymphocytes (TBL3) or *T. parva*-infected (TpM) lymphocytes. **b** Assessment of parasite viability using quantitative RT-qPCR analysis of parasite *TaSP* expression relative to bovine *actin* mRNA expression (host or parasite targeted first by the drug) in the three different cell lines compared to DMSO or Buparvaquone (Bup) treatment. Mean values ± SD. One-way Anova statistical test, Dunnett's multiple comparison test: ****$p < 0.0001$.

affects were associated with reduced expression of a number of bovine genes linked to transformation, including *MMP9*, *miR-155* and *HK2*. Despite the similarity in the effects of MMV668754 and Buparvaquone in all the functional assays, we show that their modes of action are likely to be distinct; MMV668754 did not inhibit the catalytic activity of the TaPin1 prolyl isomerase which was reported to be mutated in some Buparvaquone-resistant parasites[6,7]. We generated Buparvaquone-resistant cells that had no mutations in the *TaPin1* gene [Supplementary Fig. 1] but contained mutations in the parasite *CytB* gene [Fig. 8a]. Furthermore, mutations in the parasite *CytB* gene associated with Buparvaquone-resistance did not generate resistance to MMV668754/Trifloxystrobin. Modeling predictions suggest that this could be because they bind differently to the q0 site[37]. We built a homology model of the TaCytB protein (based on the known chicken cytochrome B structure[38]) and we computationally docked Buparvaquone or MMV668754/Trifloxystrobin drugs in the q0 site of the enzyme to estimate the predicted binding energy of each compound [Supplementary Fig. 2]. Hypothetically, MMV668754/Trifloxystrobin might target TaCytB, even in cells with Buparvaquone-resistant mutations. It is also possible that the MMV668754/Trifloxystrobin drug has distinct targets in infected cells that have not yet been identified. MMV668754 is the drug Trifloxystrobin which is a broad-spectrum fungicide used against many fungal pathogens in pest control for grapes, apples and peanuts[36]. Trifloxystrobin is thought to work by interfering with the mitochondrial respiration pathway and Cytochrome b action[37]. Interestingly, field samples from drug-resistant parasites include *TaPin1* and/or *CytB* mutations;[7] around 40% of these

field isolates had mutations in the *TaCytB* gene without any *TaPin1* mutations. Thus, mutations in either, or both, of these genes is associated with most reported cases of resistance. We cannot rule out the possibility that both drugs also target other parasite or host proteins. Combined MMV668754 and Buparvaquone drug protocols might offer reduced chances of double drug-resistance. Investigating the mechanisms and molecular targeting by MMV668754 will be of interest for further studies and could shed light on how *Theileria* spp. hijack host cell pathways. Future studies to generate crystal structures of TaCytB bound to Buparvaquone or Trifloxystrobin will provide important insights.

The MMV Pathogen Box is a rich resource of compounds tested against a wide range of pathogens[4]. Several screens of this library for anti-*Theileria* compounds have reported different sets of potential hit compounds with relatively little overlap[4,26–28]. This may result from the different screening strategies;for example, use of the MMV Pathogen Box library in this study or the MMV Malaria Box in others[4,27], focusing mostly on host cell cytotoxicity. The different MMV screening studies identified surprisingly few overlapping compounds. This could be due to the parasite species, the host cell type or the sensitivities of the screening approaches. As shown in Fig. 1c, our experimental strategy identified several compounds that overlap with those identified in other screens. Our series of subsequent assays, including quantification of the parasite load per host cell, comparing matched infect vs uninfected cells lines and testing against different *Theileria* species in both macrophages and lymphocytes, allowed us to focus on a single compound that is a promising new

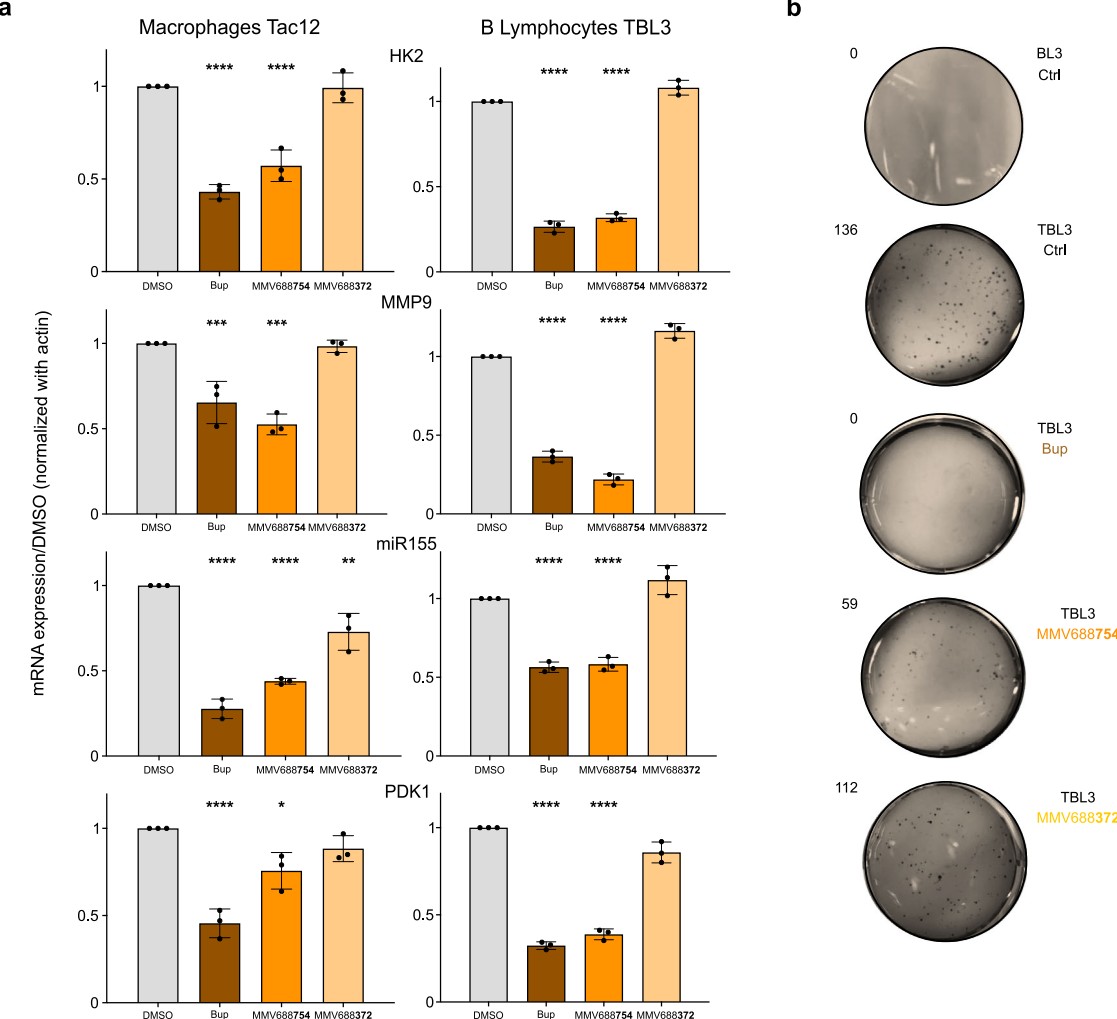

**Fig. 5 MMV668754/Trifloxystrobin inhibited the transformed phenotype. a** Infected macrophages (TaC12) or infected B lymphocytes (TBL3) were treated with the designated drugs MMV688754, MMV688372 or Buparvaquone (Bup) for 48 h and key transformation associated genes were evaluated by qRT-PCR analysis compared to DMSO controls. The MMV668754/Trifloxystrobin and Bup treatment had equivalent inhibitory effects on bovine gene expression. Mean values ± SD, standardized using bovine actin expression. $n = 3$ independent experiments. One-way Anova statistical test, Dunnett's multiple comparison test: *$p < 0.05$, **$p < 0.01$, ***$p < 0.001$, ****$p < 0.0001$. **b** Soft-agar analysis demonstrated that the drug treatment also affected the colony growth of TBL3 cells. The results shown are representative of two independent experiments.

potential drug. Future studies will be required to identify the drug target (such as our example with TaPin1 and Buparvaquone) and generate chemical derivatives for drug optimization before in vivo validation. We cannot definitively prove the targets for these anti-*Theileria* drugs in the absence of powerful genetic tools to generate mutant and rescued parasite strains. However, the finding that MMV668754/Trifloxystrobin does not inhibit TaPin1 activity and appears to be effective against Buparvaquone-resistant CytB mutants suggest that it may be an effective alternative to Buparvaquone and a promising candidate to treat cases of emerging drug-resistance.

## Methods

**Cell culture and drug screening**. All bovine cell lines were previously documented: TBL3 cells were derived from in vitro infection of the spontaneous bovine B lymphosarcoma cell line, BL3, with Hissar stock of *T. annulata*. TaC12 is a line of *T. annulata*-infected bovine macrophages. The TpMD409 lymphocyte cell line is infected with *T. parva*. Cells were cultured in RPMI 1640 (Gibco-BRL), supplemented with 10% heat-inactivated fetal bovine serum (FBS), 4 mM L-glutamine, 25 mM HEPES, 10 mM β-mercaptoethanol and 100 mg/ml of penicillin/streptomycin in a humidified 5% $CO_2$ atmosphere at 37 °C. For screening, TaC12 infected macrophages expressing a GFP-CLASP fusion protein were plated in 96-well plates and treated with the MMV compounds (at 10 μM, 2 μM or 0.4 μM concentrations)

for 48 h. Immunofluorescence was performed on fixed cells using a specific anti-H3K18me1 antibody to label the parasite nuclei. Cells were incubated with DAPI to detect host and parasite nuclear DNA. The parasite surface membrane was detected using GFP-CLASP fluorescence. Image capture (30 fields per condition) and analysis was performed with the Opera Phenix microscope (Perkin Elmer, Photonic BioImaging platform, Pasteur Institute) and the associated Acapella Software to monitor host and parasite survival [See Fig. 1a]. TBL3-Buparvaquone-resistant infected cells were generated by cultivating TBL3 cells for 70 days in the presence of Buparvaquone at 200 ng/ml (Chemos GmbH, Ref: 88426-33-9). Cells died massively during the first 192 h and the proliferation was reduced up to day 30 in culture, and then cells started to proliferate again.

**Assays for cell viability, proliferation and cell cycle analysis**. To measure cell viability, $1 \times 10^4$ cells were plated in 96-well plates in triplicate before MMV drug treatments for 48 h. Cell viability was measured using the Cell Proliferation Kit II–XTT (Roche) and the GloMax-Multi Detection System (Promega) following manufacturers' instructions. We used a two-layer soft-agar culture system to assay for colony formation and proliferation. $2 \times 10^4$ bovine cells were plated in 1.5 ml agarose (0.7% SeaKem ME Agarose; Lonza, ref: 50011, plus 2x DMEM 20% fetal calf Serum) over a 1.5-ml base layer (1% SeaKem ME Agarose plus 2X DMEM 20% fetal calf Serum) in 6-well plates. Cultures were incubated in humidified 37 °C incubators with 5% $CO_2$ in air and monitored for growth using a microscope. At the time of maximum colony formation (10–15 days in culture), final colony numbers were counted manually after fixation a staining with 0.005% Crystal Violet (Sigma, ref: C3886). For cell cycle analysis, cells at $1 \times 10^4$/ml were stained

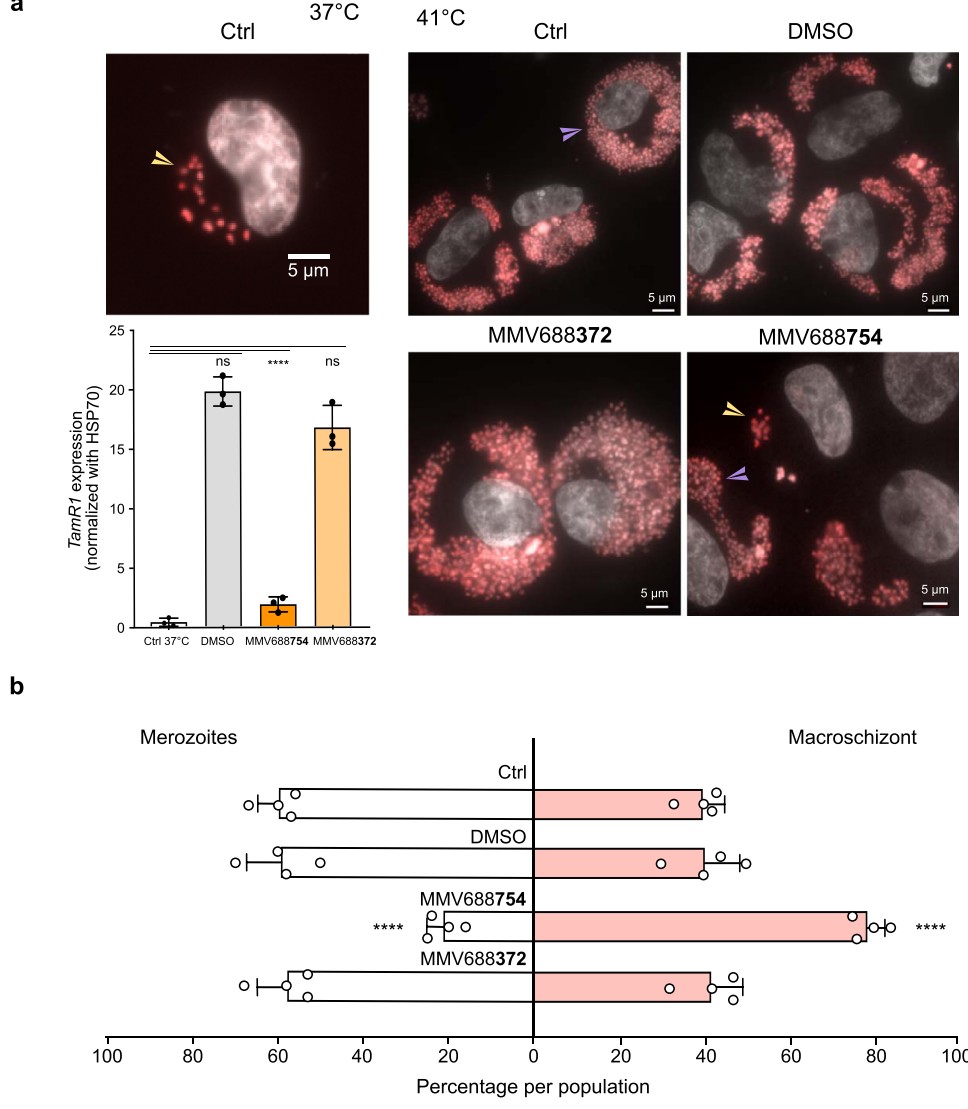

**Fig. 6 MMV668754/Trifloxystrobin inhibited parasite differentiation to merozoites (merogony). a** TaC12 infected macrophages were incubated at 37 °C or at 41 °C for 8–10 days to induce merogony. Immunofluorescence with an antibody against H3K18me1 highlighted the parasite nuclei in the macroschizont stage (yellow arrowheads) and decreases upon differentiation, when host cells fill up with large number of parasites as they differentiate into merozoites (purple arrowheads). Treatment with MMV688754 blocked the merogony process. This was confirmed by qPCR expression analysis of the parasite *TamR1* gene, a marker of the merogony stage. $n = 3$ independent experiments. One-way Anova statistical test, Dunnett's multiple comparison test. Mean values ± SD. ns = not significant, ****$p < 0.0001$. **b** Quantification of the percentage of cells at the macroschizont or merogony stage after 8–10 days in culture at 41 °C, with or without MMV688754, or DMSO treatments. $n = 4$ independent experiments ($n > 50$ cells). Mean values ± SD. One-way Anova statistical test, Dunnett's multiple comparison test. ns = not significant, ****$p < 0.001$.

with propidium iodide (PI) after different drug treatments. Cells were washed in PBS, fixed in ice-cold ethanol (70% v/v) and stored at 4 °C. For analysis, cells were washed in PBS and suspended in PI (25 µg/ml) in PBS with RNase A (200 mg/ml). Around, $10^5$ events for each sample were analysed. Flow cytometry analyses were carried out on a FACScalibur system (BD Biosciences).

**Schizont differentiation (merogony).** Macroschizont infected TaC12 cells were induced to differentiate to merogony by increasing the incubation temperature to 41 °C. Cells were passaged each time they reached confluence and $2 \times 10^6$ cells collected at day 0 (macroschizont stage) and day 8 (merogony stage) for RNA extraction and $4 \times 10^3$ cells per immunofluorescence at the same time-points.

**Immunofluorescence analysis.** Cultured *T. annulata*-infected macrophages (TaC12) were washed with PBS containing 1 mM EDTA and $3 \times 10^4$ cells per slide were centrifuged with Cytospin (10 min at 277 g) to adhere to the slide. Cells were fixed in 3.7% paraformaldehyde for 15 min and subsequently permeabilized in 0.2% Triton X-100 (prepared in PBS) for 10 min. Fixation, permeabilization and all the following steps were carried out at room-temperature. Slides were blocked with PBS 0.2% Tween (PBST)–1% BSA for 30 min. Rabbit anti-H3K18me1 (ab177253,

Abcam, 1:5000 dilution) antibodies were diluted in PBST and incubated for 1 h. Cells were subsequently washed three times with PBST and incubated with secondary antibody for 30 min with Alexa594-conjugated donkey anti-rabbit antibody (1:500 dilution). Cells were washed three times with PBST and finally, mounted on coverslips adding ProLong Diamond Antifade Mountant implemented with DAPI counterstain (Thermo Fischer Scientific). Samples were analysed using a Leica DMI6000 epifluorescence microscope. Images were generated and processed using Metamorph and ImageJ software. Parasite load was quantified by counting parasite nuclei in the host cytoplasm by DAPI. The macroschizont or merogony stages were quantified using ImageJ. We defined a threshold for the Schizont/Merogony cycle stage of 50 parasites per host cell.

**RNA extraction and RT-qPCR.** Total RNA was extracted using a Nucleospin RNA extraction kit (MachereyNagel) following the manu-facture's protocol. 1 µg of total RNA was reverse transcribed with Superscript III Reverse transcriptase Kit (Invitrogen). Real-time quantitative PCR was performed to analyze relative gene expression levels using SyberGreen Master Mix (Applied Biosystem) following the manufacturer's protocol. Relative expression values were normalized with housekeeping gene mRNA *HSP70* or bovine *Actin*. The primer sequences used are listed in Supplementary Table 2.

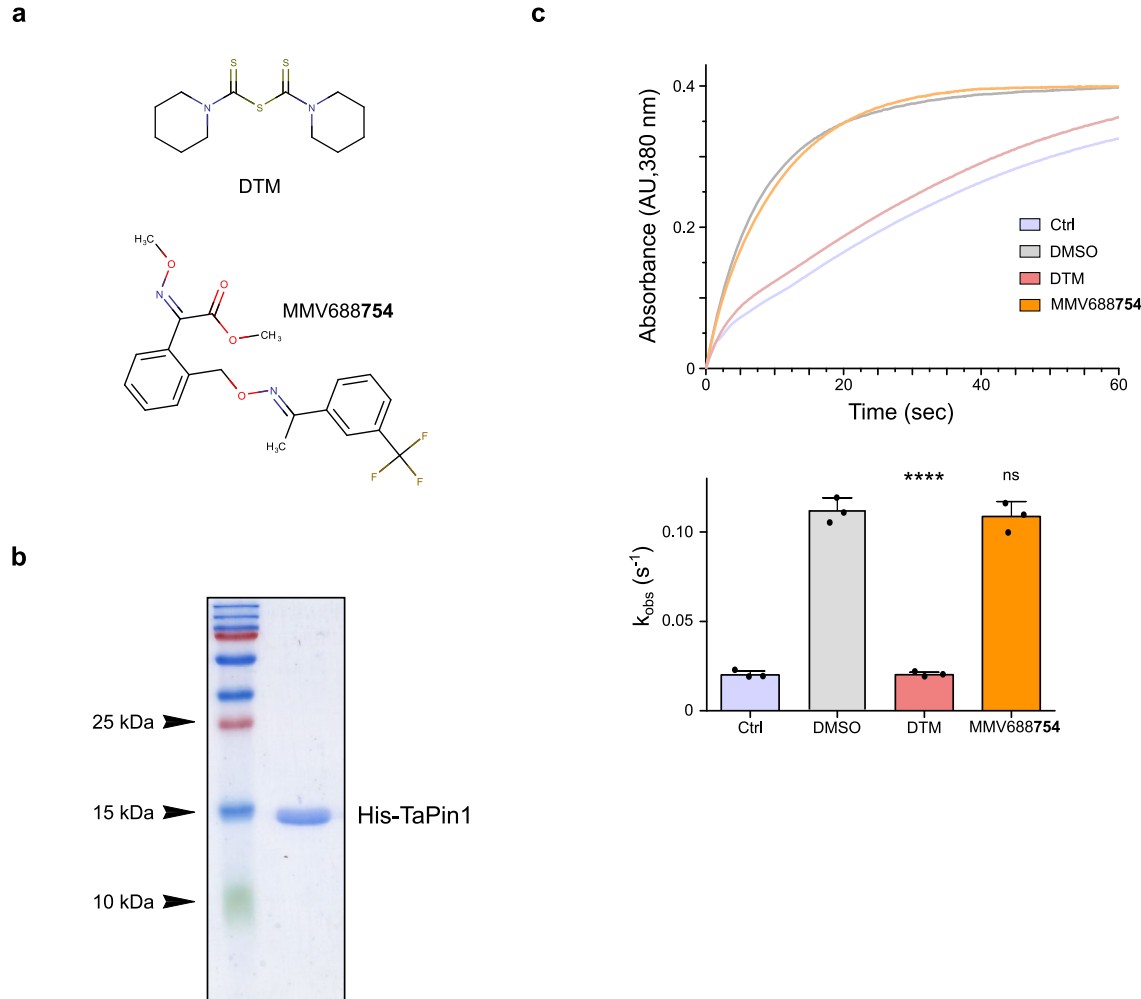

**Fig. 7 MMV688754 does not inhibit TaPin1 activity. a** Chemical structures of DTM (Pin1 inhibitor) and the test compound MMV688754. **b** Purification of recombinant His-tagged TaPin1 protein. **c** Inhibition assay showing that DTM alone inhibited TaPin1 prolyl-isomerase activity in vitro. TaPin1 (10 μM) was incubated for 24 h at 4 °C degrees with 500 μM DTM, MMV688754 or DMSO. Mixtures were diluted 1000-fold, and TaPin1 activity was assayed for 1 min by reading the absorbance at 380 nm. Apparent velocity constants ($k_{obs}$) were extrapolated from these curves. While DTM effectively inhibited TaPin1 activity, MMV688754 did not have any inhibitory effect. These results are representative of three independent experiments, which are shown quantitatively below. Mean values ± SD. $n = 3$ independent experiments. One-way Anova statistical test followed by Dunnett's multiple comparison test: ****$p < 0.0001$.

**Analysis of TaPin1 Prolyl isomerase activity**. Complementary DNA (cDNA) coding for the short form (amino acids 19–145) of *T. annulata* TaPin1 protein was amplified from TBL3 cells global cDNA and subcloned into pET28a plasmid using NdeI/XhoI restriction enzymes. BL21 HI-Control™ (DE3) *E. coli* were transformed with pET28a TaPin1 plasmid in order to express and purify 6xHis-tagged recombinant proteins. To produce recombinant TaPin1 protein, *E. coli* bacteria were cultured at 37 °C to an OD of 0.6. Protein expression was then induced by adding 500 μM iso-propyl-1-thio-β-D-galactopyranosid (IPTG) and lowering the temperature to 16 °C overnight. The bacteria were pelleted by centrifugation (2000 g, 20 min), washed with cold PBS, and harvested by centrifugation (2000 g, 15 min). Pellets were then stored at 80 °C or used directly. Bacteria were resuspended in lysis buffer (PBS 1X, 300 mM NaCl, pH 8, 1 % Triton X-100, 1 mg/ml lysozyme and protease inhibitor cocktail) and incubated for 30 min at 4 °C under agitation. Lysates were sonicated on ice (10 sec ON, 20 sec OFF, 10 min run and 20% power) and centrifuged (15,000 g, 30 min, 4 °C). The supernatant was incubated 2 h on ice in the presence of 10 mM imidazole and His-select nickel beads (Sigma). Beads were then poured into a column and washed successively with washing buffer (PBS 1X, 300 mM NaCl, pH 8) containing 0.1% Triton X-100 and washing buffer alone. Proteins were eluted in elution Buffer (PBS 1X, 300 mM NaCl, pH 8- and 300-mM imidazole) and reduced with 10 mM DTT for 20 min on ice. The purified protein was then buffer exchanged into TaPin1 buffer (35 mM HEPES, pH 8) using a PD 10 desalting column (GE Healthcare). Protein concentration was measured using the Bradford assay with BSA as standard and protein purity was assessed by SDS-PAGE analysis. Proteins were stored at −80 °C until used. To measure catalytic activity TaPin1 (10 μM) was incubated overnight at 4 °C with 500 μM DTM, MMV688754 or vehicle (DMSO). Mixtures were diluted 1000-fold (10 nM final Tapin1, 500 nM final inhibitors) and incubated with 0.2 mg/ml

chymotrypsin (Sigma, C4129) and 60 μM generic TaPin1 substrate peptide Suc-Ala-Glu-Pro-Phe-pNA (Bachem, L-1635) in cold TaPin1 buffer. Isomerization was followed by reading the absorbance of the solution at 380 nm for 1 min using a spectro-photometer (UV-1650PC, Shimadzu, France). Absorbances were then plotted against times and curves were non-linearly fitted using equation (OriginPro 8.0). $k_{obs}$ (apparent velocity constant) values were extrapolated from these fits.

**Sequencing of TaCytB mRNA in TBL3 and Buparvaquone-resistant TBL3 cell lines**. Total RNA was extracted using a Nucleospin RNA extraction kit (Macher-eyNagel) following the manu-facture's protocol. 2 μg of total RNA were reverse transcribed into cDNA with Superscript III Reverse transcriptase Kit (Invitrogen). The 1092 bp region coding for full-length TaCytB protein was amplified using TaCytB primers (Supplementary Table 2) and subcloned into pcDNA3.1 plasmid. Resulting plasmids were sequenced (Eurofins) and sequences were manually checked to correct possible base calling errors.

**Molecular docking experiments**. Homology model of TaCytB was generated with SWISS-Model[39] using the structure of the chicken Cytochrome bc1 com-plexed with an inhibitor (PDB code: 4U3F) as template. These two proteins exhibit 35% sequence identity. The modeled structure was used for molecular docking experiments to evaluate the predicted binding energy of the Buparva-quone and MMV668754/Trifloxystrobin drugs in the q0 site of the TaCytB protein. AutoDock suite[40] was used for computational docking experiments. Protein, ligands and grids were prepared using AutoDockTools (version 1.5.6). Grid maps were computed using AutoGrid (version 4). The grid points in X, Y

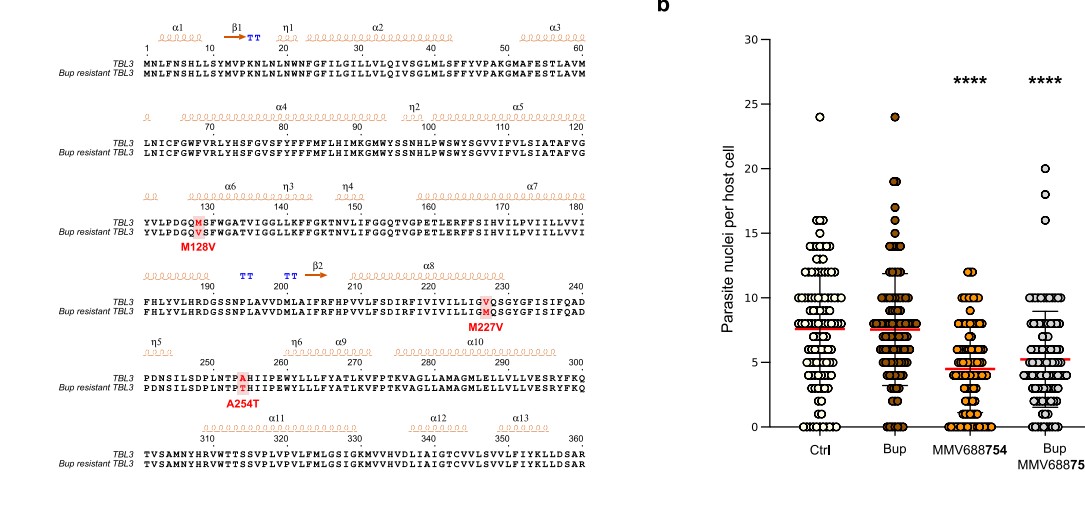

**Fig. 8 MMV668754/Trifloxystrobin and Buparvaquone may have distinct target inhibition. a** Sequence alignment of parasite TaCytB protein sequences from TBL3 cells and Buparvaquone-resistant TBL3 cell lines. Positions of the mutations are highlighted in red. **b** Parasite load per infected host cell in Buparvaquone-resistant TBL3 cell line. Parasite number was evaluated after drug treatment (2 µM for 48 h) compared to DMSO (Ctrl) or Buparvaquone (Bup) treated cells. Host cells (at least $n = 50$) were assessed for each experiment and parasite nuclei (DAPI staining) per host cell were counted manually. These results are representative of three independent experiments. Kruskal Wallis non-parametric test followed by Dunn's multiple comparison test. ****$p < 0.0001$.

and Z axis were set at $75 \times 75 \times 75$ with a grid-spacing of 0.303 Å. The grid center was centered on the q0 binding site estimated from the 4UF3 structure (PDB code). Molecular docking experiments were performed using AutoDock (version 4.2) and the Lamarckian genetic algorithm. Default protocol was applied, with an initial population of 150 randomly placed individuals, a maximum number of $2.5 \times 10^5$ energy evaluations, and a maximum number of $2.7 \times 10^4$ generations. The mutation rate and crossover were set up at 0.02 and 0.8, respectively. Fifty runs were launched using genetic algorithm searches resulting in a set of fifty docked conformations of each ligand. Docked conformations were then clustered using a RMSD threshold of 1.0 Å. The free energy of binding of the docked ligand to the receptor was predicted.

**Statistics and reproducibility**. All statistical tests used, all sample sizes, and the number of replicates are described in the corresponding figure legends. One way or two ways ANOVA test followed by Dunnet's multiple comparison tests or Kruskall-Wallis test followed by Dunn's multiple comparison tests were conducted using GraphPad (Version 9.0.0).

**Reporting summary**. Further information on research design is available in the Nature Research Reporting Summary linked to this article.

## Data availability
All data generated or analysed during this study are included in this published article (and its supplementary information files). Source data underlying the graphs are presented in Supplementary Data 1.

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

## Acknowledgements

We thank members of the Weitzman laboratory for critical reading of the manuscript and invaluable advice on this study and members of the UMR7216 for helpful discussions. We gratefully acknowledge the Medicines for Malaria Venture (MMV, Geneva, Switzerland) for providing the MMV Pathogen Box compound library. We thank G. Langsley (Institut Cochin, Paris, France) for the generous gift of TBL3, BL3 and TpMD409 cells and K. Woods (University of Bern, Switzerland) for TaC12 cells. This work was supported by the French National Research Agency (ANR PATHO-METHYLOME #ANR-15-CE12-0020), the EUR G.E.N.E. (#ANR-17-EURE-0013), and the "Who Am I?" Laboratory of Excellence #ANR-11-LABX-0071 funded by the French Government through its "Investments for the Future" program operated by the ANR under grant #ANR-11-IDEX-0005-01, the Fondation ARC pour la Recherche sur le Cancer (ARC n°228308_ARCP, AAP ARC 2020 PJA3), the PARA-SET project funded by IDEX UP AAP EMERGENCE (#IDEX-2021-I-053). J.B.W. is a senior member of the Institut Universitaire de France (IUF). We are grateful to the support of the technical platforms at the Université Paris Cité including EPI2, Epifluorescence Microscopy for Epigenetics and ImagoSeine facility, member of IBiSA and France-BioImaging infrastructure (ANR-10-INBS-04).

## Author contributions

J.B.W. developed the concept and provided overall supervision. J.B.W., S.M., F.R.L. and G.F.S. sought funding and supervised experiments. J.B.W., M.V., N.L. and J.B. designed the study, analysed the results and wrote the manuscript. N.L., S.L. and G.F.S. developed the microscopy-based screening approach. M.V., N.L., J.B. and S.L. performed the experiments. L.R. performed the structural modeling analysis

## Competing interests

The authors declare no competing interests.
