## [Peer Review File · Communications Biology]

Reviewers' comments:

Reviewer #1 (Remarks to the Author):

What are the major claims of the paper?

The authors describe the development of a microscopy-based screening strategy to identify compounds that could be effective drugs against *Theileria*. They screened compounds from the MMV Pathogen Box library for compounds with ability to kill a *T. annulata*-infected bovine macrophage cell line TaC12. This led to the identification of Trifloxystrobin (MMV688754), a broad-spectrum fungicide widely used to treat other pathogens. They went on to claim that MMV688754 blocks parasite differentiation to merozoites and does not inhibit their previously described TaPin1 enzyme.

Are they novel and will they be of interest to others in the community and the wider field?

The submission describes a 5th screen of the MMV Pathogen Box for compounds active against *Theileria*, so it's not an original idea or a novel technical approach. What is surprising is the low degree of overlap between the compounds identified (only 4 common) in the different screens (references 4, 26, 27 & 28 & in the submission). This is acknowledged by the authors and is ascribed to different screening strategies, but what's particularly surprising is that one of the published screens (ref 27) used the same TaC12 macrophage line and very similar strategies including imaging. Hostettler et al (ref 27) used electron microscopy (rather than immunofluorescence in the submission) to examine the effect of selected compounds on parasite ultrastructure. As each of the 5 screens of the MMV Pathogen Box identified different most active anti-*Theileria* drug candidates the interest of MMV688754 to others in the community and the wider field seems limited.

If the conclusions are not original, it would be helpful if you could provide relevant references.

The conclusion that MMV688754 acts independently of the "TaPin1 pathway" is based on its failure inhibit the *in vitro* prolyl isomerase activity of TaPin1, but the authors point out that "There are likely other mechanisms that contribute to the hijacking of host cell functions", so is it really surprising that MMV688754 acts independently of the "TaPin1 pathway"?

They convincingly demonstrate that MMV688754 kills *Theileria* schizonts rendering their claim that it also blocks the developmental switch from schizonts into merozoites unfounded, since dead schizonts are not capable of developing into merozoites.

Is the work convincing, and if not, what further evidence would be required to strengthen the conclusions?

From their published work it's clear that TaPin1 genes from *Theileria* parasites of different geographical origins harbor a number of independent SNPs and these often, but not always, arise in buparvaquone-resistant (TaCytB mutant) clinical isolates (see Graphical Abstract of reference 7 in the submitted manuscript). This suggests, but doesn't directly demonstrate that SNPs in TaPin1 contribute in some unknown way to buparvaquone resistance. However, what would improve the current submission is testing whether TaCytB mutant parasites display enhanced resistance to MMV688754. In the Salim et al paper (ref 7) they describe 7 taCytB mutant lines that harbor wild type TaPin1. This recommendation stems from the fact that the established buparvaquone target TaCytB (refs 9 & 10 in the submission) is mitochondrial and on line 236 the authors write "Trifloxystrobin is thought to work by interfering with the mitochondrial respiration pathway". Although this statement wasn't referenced there are indeed several publications to support that similar to buparvaquone Trifloxystrobin targets mitochondrial respiration (PMID: 33582629; PMID: 33548355; PMID: 32721740; PMID: 31344535; PMID: 27853101).

On a more subjective note, do you feel that the paper will influence thinking in the field?

No more and probably less than the 4 published anti-Theileria screens of the MMV Pathogen Box.

Additional Remarks:

Given that they only show that MMV688754 does not inhibit the in vitro prolyl isomerase activity of TaPin1 it seems a gross exaggeration that both the title and abstract state that MMV668754/Trifloxystrobin acts independently of the TaPin1 effector, or the parasite TaPin1 prolyl isomerase pathway, as it could inhibit, or be independent of many other targets/pathways such as, but not only, TaCytB.

As they clearly demonstrated that MMV668754/Trifloxystrobin efficiently kills schizonts it also seems an exaggeration to write in the Abstract (line 36) that "Trifloxystrobin also inhibited parasite differentiation to merozoites (merogony)".

Reviewer #2 (Remarks to the Author):

Highly interesting and informative study, which is well written, and shows the usefulness of drug repurposing for identifying promising compounds against Theileria infection.

Using an innovative screening procedure employing the MMV Pathogen box, the authors have identified compounds that exhibits profound in vitro activity against Theileria annulata, but also T. parva. The compound Trifloxystrobin exhibited profound activity on parasite viability rather than on the host cell. The work is clearly novel, original, and of interest for a wider community.

Interestingly, the screen employed here did not pick up buparvaquone, the drug most widely used for treatment of theileriosis. While this is stated in the manuscript, it leaves the feeling that the screen will not reflect well what could happen in vivo. A comment on this should be provided. It would be interesting to get more information on the potential in vivo applicability of this drug. Such as whether the concentrations applied in vitro would more or less correspond to the exposure that could be achieved in vivo. What are the PK parameters of this compound?

Figure 1 and associated comments: the screening procedure is nicely outlined. The authors remark on the lack of overlap with other studies with respect to compounds that were identified to have activity against Theileria. I noted, however, that the Hostettler et al and Van Voorhis et al studies report on the screening of the MMV MALARIA box, while the screen in this study concerns the MMV PATHOGEN box. I guess these corresponding statements should be clarified. Also note that "Hostettler 2017" indicated in Figure 1C is wrong, it should read "Hostettler 2016".

Have the authors considered to look at the previously identified compounds (e.g. those from the malaria box) using their screening methodology? Then a comparison to these previous studies would make more sense

The authors have previously shown that buparvaquone, besides targeting cytochrome b, also targets TaPin1 propyl isomerase. Thus, the two main known drug targets in Theileria are cytochrome b and TaPin1. Here they show convincingly that trifloxystrobin does not interfere in TaPin1 activity. What about cytochrome b? Trifloxystrobin is a Qo site inhibitor, thus interferes in mitochondrial oxidative phosphorylation (as buparvaquone). It would be important to know whether buparvaquone-resistant Theileria are affected by Trifloxystrobin or not.

Reviewer #3 (Remarks to the Author):

The manuscript "Trifloxystrobin blocks the growth and differentiation of Theileria parasites independently of the TaPin1 effector" of Villares et al., describes the result of a screen for a novel drug against the macroschizont infected cell of the tick-borne parasite Theileria annulata. Drugs from the MMV Pathogen tool box were used and results assessed using a microscope, based assay for effect on parasite viability (nuclear load per cell).

The manuscript was very well written, and the data well-presented and clear. In my assessment, there is no doubt that the paper has identified a drug that is comparable in efficacy to the current standard drug: buparvaquone (Bpq).

I do have a bit of a concern though, that trifloxystrobin may not be sufficiently distinct from buparvaquone for it to be a real game changer.

The data indicating that its mode of action is distinct from inhibition of the parasite prolyl isomerase, TaPIN1 (a target of Bpq) in vitro is convincing; but the issue concerns the known mode of action of trifloxystrobin. A quick google search provided the following:

"The strobilurin, or QoI fungicides (FRAC group 11) are extremely useful in controlling a broad spectrum of common vegetable pathogens. You may know some of older strobilurins as azoxystrobin (Quadris), trifloxystrobin (Flint). All strobilurin fungicides inhibit fungal respiration by binding to the cytochrome b complex III at the Q0 site in mitochondrial respiration. Simply said, the fungicide works by inhibiting the fungi's ability undergo normal respiration. The strobilurin chemistries have a very specific target site, or mode-of-action (MOA).

Although highly effective, fungicide chemistries like those in FRAC group 11, with a very specific MOA, are susceptible to fungicide resistance development by some fungi. Why is that? In the strobilurin's, a single nucleotide polymorphism of the cytochrome b gene leads to an amino acid substitution of glycine with alanine at position 143 of the cytochrome b protein. <https://plant-pest-advisory.rutgers.edu/understanding-the-strobilurin-fungicides-frac-group-11-2015/>

Therefore, based on the above and on the fact that buparvaquone is thought to operate by inhibiting cytochrome b (in addition to TaPIN1), one could have the view that trifloxystrobin is operating like buparvaquone and that the issue with the plentiful mutations in the Theileria cytochrome b gene could also generate resistance to trifloxystrobin. Thus, the implication in the paper that the identified drug is independent of the action of buparvaquone and is a novel drug that will get around all resistance to bpq is possibly misleading.

Therefore, in my view the authors should detail why and how trifloxystrobin MOA is independent of Bpq - is the structure of the two drugs significantly different? are they binding to a different site of cytochrome b; are there potential problems with resistance? And ideally, if possible, the authors should test whether trifloxystrobin can kill the parasite in a Bpq resistant infected cell line (with known mutation in Cytochrome b and TaPIN1 genes). These lines have been isolated and should be available via Turkish or Tunisian researchers and while a mutant TaPIN1 parasite might not be represented, a mutant Cytochrome b should be.

Minor points

Line 50: Should be "tropical theileriosis", also present in southern Europe and North Africa

Line 61: T. annulata primarily infects myeloid cells rather macrophages per se

Line 94: Should be "parasite nuclei number per....."

Line 95: Should be "we expected that compounds which kill the parasite would also result in....."

Fig 1A: Should be "Add compounds from the..." And "TaC12" not Tac12 – correct throughout manuscript

Line 101: Why do different studies pick out different drugs – is it cell line related or culture conditions – if cell line is it related to differences in parasite or host genotype/phenotype (discussion indicates host)

Line 121: Explain what mean by general stress response – simplest explanation is that it effects host more than parasite and could impact non-infected cells – ie drug targets host cell molecule.

Line 119: Says Trifloxystrobin is more effective the Bpq in reducing parasite cell survival – but this is not apparent in Fig 2A – only infected cell survival – does that mean there is an additional effect on the host cell or a different mode of action (more rapid mode of action than Bpq)

Line 193: not clear why need to test on differentiation – if Trifloxystrobin (like Bpq) kills the schizont and reduces nuclear number per parasite – differentiation is not likely to occur efficiently. If on the basis of preventing transmissible forms - might have been interesting to know effect of Bpq on differentiation – better or not than Trofloxystrobin

Line 225: what assay was used to indicate apoptosis?

Line 278: should be "were washed in PBS..."

Line 295: does not make sense and no dilution for anti-Rhoptry Ab given

Line 302: "Parasite load" is more accurate than "Parasite survival"

Line 314: centrifugation is indicated by rpm or xg – should use one or the other, preferable xg

Line 334: vehicle should be "vehicle"

Line 575: should be "compared"

Communications Biology COMMSBIO-21-2654A**Trifloxystrobin blocks the growth and differentiation of *Theileria* parasites independently of the TaPin1 effector****Reviewers' comments:**

We are grateful to the three reviewers for their remarks and their insightful suggestions. The reviewers' comments encouraged us to perform new experiments and to add new data to the manuscript. The additional Figure 8 shows our experiments using newly-generated Buparvaquone-resistant strains. Although Buparvaquone resistance has been documented in parasites from the field, we are not aware of any previous experiments with drug-resistant cells in the laboratory or studies to investigate TaPin1 and TaCytB together. Our new data show that Buparvaquone-resistant cells harbor TaCytB mutations described in the field, that they are still susceptible to the Trifloxystrobin and that the two drugs have distinct binding affinities for TaCytB. We feel that these data strengthen our study and our conclusion that Trifloxystrobin could be a promising drug to treat Buparvaquone-resistant parasites. We have changed the title and made modifications to the text to answer the reviewers' comments.

Reviewer #1 (Remarks to the Author):

What are the major claims of the paper?

The authors describe the development of a microscopy-based screening strategy to identify compounds that could be effective drugs against *Theileria*. They screened compounds from the MMV Pathogen Box library for compounds with ability to kill a *T. annulata*-infected bovine macrophage cell line TaC12. This led to the identification of Trifloxystrobin (MMV688754), a broad-spectrum fungicide widely used to treat other pathogens. They went on to claim that MMV688754 blocks parasite differentiation to merozoites and does not inhibit their previously described TaPin1 enzyme.

Are they novel and will they be of interest to others in the community and the wider field?

The submission describes a 5th screen of the MMV Pathogen Box for compounds active against *Theileria*, so it's not an original idea or a novel technical approach. What is surprising is the low degree of overlap between the compounds identified (only 4 common) in the different screens (references 4, 26, 27 & 28 & in the submission). This is acknowledged by the authors and is ascribed to different screening strategies, but what's particularly surprising is that one of the published screens (ref 27) used the same TaC12 macrophage line and very similar strategies including imaging. Hostettler et al (ref 27) used electron microscopy (rather than immunofluorescence in the submission) to examine the effect of selected compounds on parasite ultrastructure. As each of the 5 screens of the MMV Pathogen Box identified different most active anti-*Theileria* drug candidates the interest of MMV688754 to others in the community and the wider field seems limited.

If the conclusions are not original, it would be helpful if you could provide relevant references.

The conclusion that MMV688754 acts independently of the "TaPin1 pathway" is based on its failure inhibit the in vitro prolyl isomerase activity of TaPin1, but the authors point out that "There are likely other mechanisms that contribute to the hijacking of host cell functions", so is it really surprising that MMV688754 acts independently of the "TaPin1 pathway"?

We have added new data and a new Figure 8 to demonstrate that Buparvaquone and Trifloxystrobin target TaCytB in distinct ways and showing that Buparvaquone-resistant cells are still susceptible to treatment with Trifloxystrobin. We feel that this strengthens our conclusions and adds important novel data.

They convincingly demonstrate that MMV688754 kills *Theileria* schizonts rendering their claim that it also blocks the developmental switch from schizonts into merozoites unfounded, since dead schizonts are not capable of developing into merozoites.

We cannot show that MMV688754 kills Theileria parasites, rather that it decreases parasite load. Some Trifloxystrobin-treated cells still contain parasites and the host cells did not die, but we showed that these have reduced merogony. This raises the promise as a drug that reduces schizont parasite load and inhibits differentiation, thereby decreasing the chances of transmission. We failed to observe a sufficient number of healthy cells and parasite numbers in Buparvaquone-treated cells in order to assess differentiation – this could be because it targets TaCytB and TaPin1.

On a more subjective note, do you feel that the paper will influence thinking in the field? No more and probably less than the 4 published anti-*Theileria* screens of the MMV Pathogen Box.

We hope that the new data and new Figure strengthen our study and add important novel data.

Additional Remarks:

Given that they only show that MMV688754 does not inhibit the in vitro prolyl isomerase activity of TaPin1 it seems a gross exaggeration that both the title and abstract state that MMV688754/Trifloxystrobin acts independently of the TaPin1 effector, or the parasite TaPin1 prolyl isomerase pathway, as it could inhibit, or be independent of many other targets/pathways such as, but not only, TaCytB.

Our new data and new Figure further argue for distinct modes of action on both TaPin1 and TaCytB. We cannot rule out that either of these drugs have additional targets. We have added a comment to mention this on page 9. We have modified the title in response to the reviewer's comments and our new data.

As they clearly demonstrated that MMV688754/Trifloxystrobin efficiently kills schizonts it also seems an exaggeration to write in the Abstract (line 36) that “Trifloxystrobin also inhibited parasite differentiation to merozoites (merogony)”.

See above comments.

Reviewer #2 (Remarks to the Author):

Highly interesting and informative study, which is well written, and shows the usefulness of drug repurposing for identifying promising compounds against *Theileria* infection.

We are glad that the reviewer found our study interesting, informative and well-written.

Using an innovative screening procedure employing the MMV Pathogen box, the authors have identified compounds that exhibits profound in vitro activity against *Theileria annulata*, but also *T. parva*. The compound Trifloxystrobin exhibited profound activity on parasite viability rather than on the host cell. The work is clearly novel, original, and of interest for a wider community.

We are grateful to the reviewer for commenting that our work is original and of broad interest.

Interestingly, the screen employed here did not pick up buparvaquone, the drug most widely used for treatment of theileriosis. While this is stated in the manuscript, it leaves the feeling that the screen will not reflect well what could happen in vivo. A comment on this should be provided. It would be interesting to get more information on the potential in vivo applicability of this drug. Such as whether

the concentrations applied in vitro would more or less correspond to the exposure that could be achieved in vivo. What are the PK parameters of this compound?

Figure 1 and associated comments: the screening procedure is nicely outlined. The authors remark on the lack of overlap with other studies with respect to compounds that were identified to have activity against *Theileria*. I noted, however, that the Hostettler et al and Van Voorhis et al studies report on the screening of the MMV MALARIA box, while the screen in this study concerns the MMV PATHOGEN box. I guess these corresponding statements should be clarified. Also note that "Hostettler 2017" indicated in Figure 1C is wrong, it should read "Hostettler 2016".

We have added a comment on page 10 clarifying that the previous Hostettler et al and Van Voorhis et al studies using the MMV Malaria Box library.

We have corrected the typo mistake in Figure 1C to "Hostettler 2016".

Have the authors considered to look at the previously identified compounds (e.g. those from the malaria box) using their screening methodology? Then a comparison to these previous studies would make more sense

No, we have not specifically addressed this issue. Indeed, the MMV Malaria Box is no longer available for distribution.

The authors have previously shown that buparvaquone, besides targeting cytochrome b, also targets TaPin1 propyl isomerase. Thus, the two main known drug targets in *Theileria* are cytochrome b and TaPin1. Here they show convincingly that trifloxystrobin does not interfere in TaPin1 activity. What about cytochrome b? Trifloxystrobin is a Qo site inhibitor, thus interferes in mitochondrial oxidative phosphorylation (as buparvaquone). It would be important to know whether buparvaquone-resistant *Theileria* are affected by Trifloxystrobin or not.

We thank the reviewer for raising this important issue. We have now added new experiments to address this question. We were able to generate Buparvaquone-resistant cells by prolonged growth of TBL3 cells in the presence of the Buparvaquone drug. We sequenced the TaPin1 and TaCytB genes in these cells and identified mutations in the TaCytB gene. We showed that these Buparvaquone-resistant cells are still sensitive to Trifloxystrobin treatment, suggesting that they are independent. We also performed modeling with the TaCytB protein and the two drugs to show that they have different binding affinities and docking binding modes. These results (shown in the new Figure 8) support our conclusion that Buparvaquone and Trifloxystrobin are complementary drugs and that Trifloxystrobin could be used to overcome Buparvaquone resistance.

Reviewer #3 (Remarks to the Author):

The manuscript "Trifloxystrobin blocks the growth and differentiation of *Theileria* parasites independently of the TaPin1 effector" of Villares et al., describes the result of a screen for a novel drug against the macroschizont infected cell of the tick-borne parasite *Theileria annulata*. Drugs from the MMV Pathogen tool box were used and results assessed using a microscope, based assay for effect on parasite viability (nuclear load per cell).

The manuscript was very well written, and the data well-presented and clear. In my assessment, there is no doubt that the paper has identified a drug that is comparable in efficacy to the current standard drug: buparvaquone (Bpq).

We are glad that the reviewer found our study clear and well-written.

I do have a bit of a concern though, that trifloxystrobin may not be sufficiently distinct from buparvaquone for it to be a real game changer.

We added new data and a new figure to strengthen our claim that Trifloxystrobin and Buparvaquone have distinct actions and complementary actions.

The data indicating that its mode of action is distinct from inhibition of the parasite prolyl isomerase, TaPIN1 (a target of Bpq) in vitro is convincing; but the issue concerns the known mode of action of trifloxystrobin. A quick google search provided the following:

"The strobilurin, or QoI fungicides (FRAC group 11) are extremely useful in controlling a broad spectrum of common vegetable pathogens. You may know some of older strobilurins as azoxystrobin (Quadris), trifloxystrobin (Flint). All strobilurin fungicides inhibit fungal respiration by binding to the cytochrome b complex III at the QO site in mitochondrial respiration. Simply said, the fungicide works by inhibiting the fungi's ability undergo normal respiration. The strobilurin chemistries have a very specific target site, or mode-of-action (MOA).

Although highly effective, fungicide chemistries like those in FRAC group 11, with a very specific MOA, are susceptible to fungicide resistance development by some fungi. Why is that? In the strobilurin's, a single nucleotide polymorphism of the cytochrome b gene leads to an amino acid substitution of glycine with alanine at position 143 of the cytochrome b protein.

Therefore, based on the above and on the fact that buparvaquone is thought to operate by inhibiting cytochrome b (in addition to TaPIN1), one could have the view that trifloxystrobin is operating like buparvaquone and that the issue with the plentiful mutations in the Theileria cytochrome b gene could also generate resistance to trifloxystrobin. Thus, the implication in the paper that the identified drug is independent of the action of buparvaquone and is a novel drug that will get around all resistance to bpq is possibly misleading.

Therefore, in my view the authors should detail why and how trifloxystrobin MOA is independent of Bpq - is the structure of the two drugs significantly different? are they binding to a different site of cytochrome b; are there potential problems with resistance? And ideally, if possible, the authors should test whether trifloxystrobin can kill the parasite in a Bpq resistant infected cell line (with known mutation in Cytochrome b and TaPIN1 genes). These lines have been isolated and should be available via Turkish or Tunisian researchers and while a mutant TaPIN1 parasite might not be represented, a mutant Cytochrome b should be.

We thank the reviewer for raising this important issue. We have now added new experiments to address this question.

The reports of Buparvaquone-resistant strains (Turkish, Tunisian and others) suggested that TaPin1 and TaCytB may be drug targets. However, these reports did not describe drug-resistant cell lines for further study. To address this issue we generated Buparvaquone-resistant cells by prolonged growth of TBL3 cells in the presence of the Buparvaquone drug. We sequenced the TaPin1 and TaCytB genes in these cells and identified mutations in the TaCytB gene (similar to those previously described). We showed that these Buparvaquone-resistant cells are still sensitive to Trifloxystrobin treatment, suggesting that they are independent. We also performed modeling with the TaCytB protein and the two drugs to show that they have different binding affinities and docking binding modes. These results support our conclusion that Buparvaquone and Trifloxystrobin are complementary drugs and that Trifloxystrobin could be used to overcome Buparvaquone resistance. These data are shown in the new Figure 8.

Minor

points

Line 50: Should be "tropical theileriosis", also present in southern Europe and North Africa

We added this suggestion to lines 57-58.

Line 61: *T. annulata* primarily infects myeloid cells rather macrophages per se

We modified the text of line 68

Line 94: Should be “parasite nuclei number per.....”

We inserted the word ‘nuclei’ in line 103.

Line 95: Should be “we expected that compounds which kill the parasite would also result in.....”

We made the suggest modification to the text in line 104.

Fig 1A: Should be “Add compounds from the....” And “TaC12” not Tac12 – correct throughout manuscript

We corrected throughout the manuscript.

Line 101: Why do different studies pick out different drugs – is it cell line related or culture conditions – if cell line is it related to differences in parasite or host genotype/phenotype (discussion indicates host)

We commented on this in the last paragraph of the Discussion.

Line 121: Explain what mean by general stress response – simplest explanation is that it effects host more than parasite and could impact non-infected cells – ie drug targets host cell molecule.

We cannot conclude whether the drugs target parasite or host proteins. We think that some drugs may induce signals that cause the parasites to begin to differentiate but we have no clear data to support this. We have seen similar increases in parasite numbers in other drugs screens (data not shown). We could not reproduce this enhanced parasite load at lower drug concentrations. That is why referred to this as “general stress response”.

Line 119: Says Trifloxystrobin is more effective the Bpq in reducing parasite cell survival – but this is not apparent in Fig 2A – only infected cell survival – does that mean there is an additional effect on the host cell or a different mode of action (more rapid mode of action than Bpq).

Our drug modeling data on TaCytB (presented in the new Figure 8) suggest that Trifloxystrobin may be more effective inhibitor of Cytochrome B than Buparvaquone and the experiments with TaPin1 suggest that Buparvaquone functions differently than Trifloxystrobin. We cannot rule out the possibility that both drugs have additional (parasite or host) target proteins (we commented on this on line 285)

Line 193: not clear why need to test on differentiation – if Trifloxystrobin (like Bpq) kills the schizont and reduces nuclear number per parasite – differentiation is not likely to occur efficiently. If on the basis of preventing transmissible forms - might have been interesting to know effect of Bpq on differentiation – better or not than Trofloxystrobin

We cannot conclude that Trifloxystrobin or Buparvaquone ‘kill the schizont’, rather that they reduce the number of parasite nuclei per cell. We were still able to investigate merogony in Trifloxystrobin-treated cells where some parasites were still present and the host cells did not die. This raises the promise as a drug that reduces schizont parasite load and inhibits differentiation thereby decreasing the chances of transmission. We failed to observe a sufficient number of healthy cells and parasite numbers in Buparvaquone-treated cells in order to assess differentiation – this could be because it targets TaCytB and TaPin1.

Line 225: what assay was used to indicate apoptosis?

Apoptosis was determined by flow cytometry analysis and quantification of sub-G1 populations.

Line 278: should be “were washed in PBS...”

We corrected the text on line 346.

Line 295: does not make sense and no dilution for anti-Rhoptry Ab given

We corrected the text on line 364.

Line 302: “Parasite load” is more accurate than “Parasite survival”

We modified the text at line 371.

Line 314: centrifugation is indicated by rpm or xg – should use one or the other, preferable xg

We modified the text accordingly.

Line 334: vehicule should be “vehicle

We corrected the French spelling at line 403.

Line 575: should be “compared”

We corrected this.

Reviewers' comments:

Reviewer #1 (Remarks to the Author):

Criticism of Rebuttal

We have added new data and a new Figure 8 to demonstrate that Buparvaquone and Trifloxystrobin target TaCytB in distinct ways and showing that Buparvaquone-resistant cells are still susceptible to treatment with Trifloxystrobin. We feel that this strengthens our conclusions and adds important novel data.

There is no demonstration that Trifloxystrobin targets TaCytB only in silico docking of Trifloxystrobin to a hypothetical model of TaCytB. In total contrast, they present data showing that the in vitro generated buparvaquone-resistant line harboring mutations in TaCytB remains sensitive to Trifloxystrobin. The most reasonable explanation for this genetic data is that TaCytB is not a target of Trifloxystrobin. Even though the in silico docking study to a model of TaCytB is interesting, it is speculative and should be in the Discussion section (not in the Results sections), together with other possible mechanisms of action.

We failed to observe a sufficient number of healthy cells and parasite numbers in Buparvaquone-treated cells in order to assess differentiation – this could be because it targets TaCytB and TaPin1.

In the in vitro generate buparvaquone-resistant line they sequenced both TaCytB and TaPin1, but only show mutations in TaCytB. Results of TaPin1 sequencing should be shown and discussed. If no mutations were found in TaPin1, these findings contradict their previously published report that TaPin1 contributes to buparvaquone resistance (PMID: 25624101).

Our new data and new Figure further argue for distinct modes of action on both TaPin1 and TaCytB. We cannot rule out that either of these drugs have additional targets. We have added a comment to mention this on page 9. We have modified the title in response to the reviewer's comments and our new data.

To adequately address this issue, authors should generate whole genome sequence (WGS) of drug selected parasites. This would identify the target gene(s) and exclude mutations elsewhere. It is therefore very surprising that in a way similar to their generation of a buparvaquone-resistant line they did not select for a Trifloxystrobin-resistant line. Purified schizonts (published protocols exist and have been used by several Theileria labs (and one example PMID: 25077614 is from the same lab that provided them with the GFP-CLASP expressing TaC12 line) and WGS would identify which gene(s) has mutated to confer Trifloxystrobin-resistance.

The reports of Buparvaquone-resistant strains (Turkish, Tunisian and others) suggested that TaPin1 and TaCytB may be drug targets. However, these reports did not describe drug-resistant cell lines for further study. To address this issue we generated Buparvaquone-resistant cells by prolonged growth of TBL3 cells in the presence of the Buparvaquone drug. We sequenced the TaPin1 and TaCytB genes in these cells and identified mutations in the TaCytB gene (similar to those previously described). We showed that these Buparvaquone-resistant cells are still sensitive to Trifloxystrobin treatment, suggesting that they are independent. We also performed modeling with the TaCytB protein and the two drugs to show that they have different binding affinities and docking binding modes. These results support our conclusion that Buparvaquone and Trifloxystrobin are complementary drugs and that Trifloxystrobin could be used to overcome Buparvaquone resistance. These data are shown in the new Figure 8.

To resume:

The in vitro generation of a buparvaquone-resistant line is important, as it revealed that selection by

buparvaquone did not apparently generate mutations in TaPin1. By contrast and as expected, it did generate mutations in TaCytB and moreover, allowed them to demonstrate that these mutations did not confer resistance to Trifloxystrobin. To counter the mutational/genetic evidence that negates TaCytB as the target of Trifloxystrobin they propose that it binds to TaCytB with a different mode of action to buparvaquone, one that doesn't involve the described mutations. However, this proposition lacks data being purely hypothetical relying on in silico modelling of buparvaquone docking to a putative model of TaCytB.

Reviewer #2 (Remarks to the Author):

Congratulations to the authors for a well-made revision of this paper, which resulted in a highly improved manuscript. Especially the additional experiments made important contributions.

One more point:

The authors have not responded to the question concerning PK properties of Trifloxystrobin. Has this already been investigated in cattle? This would be an important aspect if one claims that it could be a useful drug against bovine Theileria infection.

Reviewer #3 (Remarks to the Author):

This revised manuscript details the identification of a new/repurposed drug effective against cells infected with the Theileria annulata parasite. Previously the reviewers recommended that the authors test the drug (trifloxystrobin) against cell lines resistant to buparvaquone (Bpq), as the known mode of action of trifloxystrobin is similar to the anti-theilerial buparvaquone. They do this by generating a resistant line through prolonged drug pressure in vitro and showing that there are mutations in the putative cytochrome b target of Bpq. They then show this line is sensitive to trifloxystrobin and conclude the site of interaction of the two drugs differs and that trifloxystrobin could be useful to combat Bpq resistance. It would appear therefore that they have answered the main comments of the reviewers, but I suggest that they could discuss in more detail potential caveats that may still exist.

Firstly - although it has never been published: Bpq lines can be generated in vitro quite easily, and not necessarily show any mutation from a drug sensitive parental cloned cell line in the cytb gene. Thus, it is possible for other mechanisms of resistance against Bpq to occur (at least in vitro). The authors could not know of this finding and so what they have written is fine, but if they aim to validate trifloxystrobin by in vivo studies in future, it is worth bearing in mind.

Secondly do they know that the TBL3 line is represented by a single parasite genotype. Most theileria lines are represented by multiple genotypes which could mean the mutations identified already exist in the sensitive cell line but were not obtained by the method employed to sequence the gene. Can the authors add some information indicating that all probable putative alleles of the gene were likely to be obtained or the parental line is clonal. The alternative being to develop a mutation specific PCR assay to show the parental line does not possess the (mutant) gene.

Lastly: unfortunately, unlike Plasmodium it is currently impossible to prove that the mutant gene confers resistance to the parasite, since stable transfection is not available. The best that can be done is to transfer a resistant parasite to a new host cell. The authors should emphasise further that the results only show an association and do not prove the resistant mechanism/drug target. In which case line 286 could be "cannot discount that both drugs target other parasite or host proteins"

The new data indicate that resistance to Bpq can be independent of a mutation in TaPin1 - does this refute previous identification of this gene as a Bpq target or is the model that there are multiple mutually exclusive targets of the drug. Could add more detail on what their current model is for buparvaquone sensitivity/resistance involving at least 2 independent genes.

Communications Biology COMMSBIO-21-2654B**Trifloxystrobin blocks the growth of *Theileria* parasites and is a promising drug to treat Buparvaquone resistance****Replies to Reviewers' comments:**

We are grateful to the three reviewers for considering our revised manuscript. We are glad that they generally agree that the manuscript is improved and that the new data add more insight. We comment on the additional issues raised in the second round of review. We have added a Supplementary Figure, new references and several clarifications in the text.

Reviewer #1 (Remarks to the Author):**Criticism of Rebuttal**

There is no demonstration that Trifloxystrobin targets TaCytB only in silico docking of Trifloxystrobin to a hypothetical model of TaCytB. In total contrast, they present data showing that the in vitro generated buparvaquone-resistant line harboring mutations in TaCytB remains sensitive to Trifloxystrobin. The most reasonable explanation for this genetic data is that TaCytB is not a target of Trifloxystrobin. Even though the in silico docking study to a model of TaCytB is interesting, it is speculative and should be in the Discussion section (not in the Results sections), together with other possible mechanisms of action.

There is a very extensive literature detailing the targeting of Cytochrome b by Trifloxystrobin in many species [for example, doi.org/10.1021/acs.chemrev.0c00712]. While it is true that our model of TaCytB-Trifloxystrobin is not based on crystallographic structure data, it is based on reports in the literature showing that Trifloxystrobin can be modeled in the binding pocket of CytB in other species [e.g. DOI 10.1002/ps.5990]. Moreover, the mutations in the TaCytB gene are distinct from those that affect Trifloxystrobin binding [DOI 10.1002/ps.5990]. We have added a comment in the Discussion [line 262] to clarify this point.

In the in vitro generate buparvaquone-resistant line they sequenced both TaCytB and TaPin1, but only show mutations in TaCytB. Results of TaPin1 sequencing should be shown and discussed. If no mutations were found in TaPin1, these findings contradict their previously published report that TaPin1 contributes to buparvaquone resistance (PMID: 25624101).

The TaPin1 sequence was identical to the published sequence, but we have added it to the Supplementary information as requested [Supplementary Fig 1] and the discussion [page 9].

*In a previous study, we reported that *Theileria* parasites from farms in Sudan with Buparvaquone resistance displayed either CytB or TaPin1 mutations [PMID: 31794951] or sometimes both. So the current study is consistent with these field data in which 40% of the TaCytB mutants do not harbor TaPin1 mutations.*

To adequately address this issue, authors should generate whole genome sequence (WGS) of drug selected parasites. This would identify the target gene(s) and exclude mutations elsewhere. It is therefore very surprising that in a way similar to their generation of a buparvaquone-resistant line they did not select for a Trifloxystrobin-resistant line. Purified schizonts (published protocols exist and have been used by several *Theileria* labs (and one example PMID: 25077614 is from the same lab that provided them with the GFP-CLASP expressing TaC12 line) and WGS would identify which gene(s) has mutated to confer Trifloxystrobin-resistance.

While we agree that Whole Genome Sequencing of the parasite (and bovine) genomes in drug-resistant cells (or parasites) would be interesting, we feel that this is beyond the scope of the current study. We have not been able to generate Trifloxystrobin-resistant parasites, but this would indeed be interesting

for future studies. The protocols mentioned enrich for the presence of schizonts but do not generate pure material. We feel these interesting experiments are also beyond the scope of the current study.

To resume:

The in vitro generation of a buparvaquone-resistant line is important, as it revealed that selection by buparvaquone did not apparently generate mutations in TaPin1. By contrast and as expected, it did generate mutations in TaCytB and moreover, allowed them to demonstrate that these mutations did not confer resistance to Trifloxystrobin. To counter the mutational/genetic evidence that negates TaCytB as the target of Trifloxystrobin they propose that it binds to TaCytB with a different mode of action to buparvaquone, one that doesn't involve the described mutations. However, this proposition lacks data being purely hypothetical relying on in silico modelling of buparvaquone docking to a putative model of TaCytB.

The reviewer appears to agree that our new data on Buparvaquone-resistant lines provided additional insights into the distinct modes of function of the Buparvaquone and Trifloxystrobin drugs. We have added a comment in the Discussion [page 9] emphasizing that these modeling data and mutational analysis will require further study.

Reviewer #2 (Remarks to the Author):

Congratulations to the authors for a well-made revision of this paper, which resulted in a highly improved manuscript. Especially the additional experiments made important contributions.

We are glad that the Reviewer found our manuscript improved and appreciated the new data.

One more point:

The authors have not responded to the question concerning PK properties of Trifloxystrobin. Has this already been investigated in cattle? This would be an important aspect if one claims that it could be a useful drug against bovine *Theileria* infection.

We added citation to a review article [DOI: 10.1002/ps.520] that includes detail discussion on the use of strobilurin fungicides .

Reviewer #3 (Remarks to the Author):

This revised manuscript details the identification of a new/repurposed drug effective against cells infected with the *Theileria annulata* parasite. Previously the reviewers recommended that the authors test the drug (trifloxystrobin) against cell lines resistant to buparvaquone (Bpq), as the known mode of action of trifloxystrobin is similar to the anti-theilerial buparvaquone. They do this by generating a resistant line through prolonged drug pressure in vitro and showing that there are mutations in the putative cytochrome b target of Bpq. They then show this line is sensitive to trifloxystrobin and conclude the site of interaction of the two drugs differs and that trifloxystrobin could be useful to combat Bpq resistance. It would appear therefore that they have answered the main comments of the reviewers, but I suggest that they could discuss in more detail potential caveats that may still exist.

We are glad that the Reviewer felt we have now answered the main comments.

Firstly - although it has never been published: Bpq lines can be generated in vitro quite easily, and not necessarily show any mutation from a drug sensitive parental cloned cell line in the cytb gene. Thus, it

is possible for other mechanisms of resistance against Bpq to occur (at least in vitro). The authors could not know of this finding and so what they have written is fine, but if they aim to validate trifloxystrobin by in vivo studies in future, it is worth bearing in mind.

Indeed, we did not find other examples of Buparvaquone-resistant cell lines in the literature. We are grateful to the Reviewer for sharing this information and will keep it in mind in future studies.

Secondly do they know that the TBL3 line is represented by a single parasite genotype. Most theileria lines are represented by multiple genotypes which could mean the mutations identified already exist in the sensitive cell line but were not obtained by the method employed to sequence the gene. Can the authors add some information indicating that all probable putative alleles of the gene were likely to be obtained or the parental line is clonal. The alternative being to develop a mutation specific PCR assay to show the parental line does not possess the (mutant) gene.

In a previous study we generated RNA-Seq data for infected (TBL3) cells. We have re-analysed these data and found no differences in the TaPin1 sequences in TBL3 cells or Buparvaquone-resistant TBL3 cells [See new Supplementary Figure 1]. Our sequencing data indeed support the suggestion that the parental line is clonal.

Lastly: unfortunately, unlike Plasmodium it is currently impossible to prove that the mutant gene confers resistance to the parasite, since stable transfection is not available. The best that can be done is to transfer a resistant parasite to a new host cell. The authors should emphasise further that the results only show an association and do not prove the resistant mechanism/drug target. In which case line 286 could be "cannot discount that both drugs target other parasite or host proteins"

We acknowledge these suggestions. We have added a cautionary note to the Discussion [page 9] and made the recommended addition to line 291.

The new data indicate that resistance to Bpq can be independent of a mutation in TaPin1 - does this refute previous identification of this gene as a Bpq target or is the model that there are multiple mutually exclusive targets of the drug. Could add more detail on what their current model is for buparvaquone sensitivity/resistance involving at least 2 independent genes.

In a previous study, we reported that Theileria parasites from farms in Sudan with Buparvaquone resistance displayed either CytB or TaPin1 mutations [PMID: 31794951] or sometimes both. So the current study is consistent with these field data in which 40% of the TaCytB mutants do not harbor TaPin1 mutations.

We added a comment to the Discussion [page 9] to clarify this point.

Reviewers' comments:

Reviewer #1 (Remarks to the Author):

Major problem:

I was very surprised to see that my very strong recommendation to move the in silico 3D-hypothetical model of TaCytB (Figure 8) out of the Results section to the Discussion section was not followed. The reason given was that a hypothetical in silico model could be useful for animating discussion, but should not be considered a result based on experimentally derived binding data. Also, I strongly recommended that the discussion should include mention of alternative potential mechanisms that might explain why *T. annulata* is sensitive to Trifloxystrobin and remains sensitive even when TaCytB is mutated and resistant to buparvaquone.

To more clearly explain why one should treat with extreme caution docking studies based on the 3D-structure derived from modelling *T. annulata* CytB on the crystal structure coordinates of chicken CytB (lines 401 & 402) comes from the quite low amino acid sequence identity (only 38%) and positivity (54%) displayed between *Theileria* CytB and chicken CytB (see below).

>Chicken CytB

```
MAPNIRKSHPLLKMINNSLIDLPAASNISAWWNFGSLLAVCLMTQILTGLLLAMHYTADTSLAFSSVAHTCRNVQYG
WLIRNLHANGASFFIFICIFLHIGRGLYGSYLYKETWNTGVILLTLMATAFVGYVLPWQMSFWGATVITNLFSAIPY
IGHTLVEWAWGGFSDNPTLTRFFALHFLLPFTIAGITIIHLTLFHESGSNNPLGISSDSDKIPFHPYYSFKDILGLTLM
LTPFLTALFSPNLLGDPENFTPANPLVTPPHIKPEWYFLFAYAILRSIPNKLGGVLALAASVLILFLIPFLHKSQRMTMTF
RPLSQTLFWLLVANLLILTWIGSQPVEHPFIIIGQMASLSYFTILLILFPTIGTLENKMLNY
```

```
>Tap370b08.q2ca38.03c-t26_1-p1 | transcript=Tap370b08.q2ca38.03c-t26_1
| gene=Tap370b08.q2ca38.03c | organism=Theileria_annulata_strain_Ankara
| gene_product=cytochrome B, putative |
transcript_product=cytochrome B, putative | location=CR940346:4452-5543(-)
| protein_length=363 | sequence_SO=mitochondrial_chromosome
| SO=protein_coding_gene | is_pseudo=false
Length=363
```

Score = 206 bits (524), Expect = 7e-64, Method: Compositional matrix adjust.
Identities = 133/351 (38%), Positives = 190/351 (54%), Gaps = 8/351 (2%)

```
Query 12 LKMINNSLIDLPAASNISAWWNFGSLLAVCLMTQILTGLLLAMHYTADTSLAFSSVAHTC 71
+ + N+ L+ P N++ WNFG +L + L+ QI++GL+L+ Y +AF S
Sbjct 1 MNLFNSHLLSYMVPKNLNLNWNFGFILGILLVLQIISGLMLSFFYVPAKGMFAFESTLAVM 60
```

```
Query 72 RNVQYGWLIRNLHANGASFFIFICIFLHIGRGLYGSYLYKETWNTGVILLTLMATAFVG 131
N+ +GW +R H+ G SF+F +FLHI +G++Y S +W +GV++ + +ATAFVG
Sbjct 61 LNICFGWFVRLYHSFGVSFYFFMFLHIMKGMWYSSNHLPSWYSGVVIFVLSIATAFVG 120
```

```
Query 132 YVLPWQMSFWGATVITNLFSAIPYIGHTLVEWAWGGFSDNPTLTRFFALHFLLPFTIA 191
YVLP GQMSFWGATVI L + + G V +GG +V TL RFF++H +LP I
Sbjct 121 YVLPDGQMSFWGATVIGGL---LKFFGKANVL-IFGGQTVGPETLERFFSIHVILPVII 176
```

```
Query 192 GITIIHLTLFHESGSNNPLGISSDSDKIPFHPYYSFKDI--LGLTLMTPFLTALFSPN 249
+ I HL LH GS+NPL + FHP F DI + + ++L + F
Sbjct 177 LVVIFHLYVLHRDGSSNPLAVIDMLAIFRFHPVVLFSDFRIVIVILLIGVQSGYGFISI 236
```

```
Query 250 LLGDPENFTPANPLVTPPHIKPEWYFLFAYAILRSIPNKLGGVLALAASVLILFLIPFLH 309
DP+N ++PL TP HI PEWY L YA L+ P K+ G+LA+A + +L L+
```

Sbjct 237 FQADPDNSILSDPLNTPAHIIPEWYLLLFYATLKVFPKTVAGLLAMAGMLELLVLLVESR 296

Query 310 KSKQ--RTMTFRPLSQTLFWLLVANLLILTWIGSQPVEHPFIIIGQMASLS 358

KQ M + + T LV L +L IG V I IG LS

Sbjct 297 YFKQTVSAMNYHRVWTTSSVPLVPVLFMLGSIGKMMVHVDLIAIGTCVVLS 347

Moreover, in spite of this low level of conserved amino acid sequence between parasite and chicken CytB, a lack of caution was not evident in the revision that liberally interchanges hypothetical "docking" for "binding" and examples in the text are listed below.

Abstract:

Line 40: Furthermore, Trifloxystrobin and Buparvaquone showed distinct binding affinities to parasite Cytochrome B.

Discussion:

Lines 259-260: Furthermore, mutations in the parasite CytB gene associated with Buparvaquone-resistance did not generate resistance to MMV668754/Trifloxystrobin, probably because they bind differently to the q0 site.

Lines 266 – 269: While our predicted structure is based on published Cytochrome b models from other species (in fact only chicken was used lines 401/402) rather than crystallographic data, the characterized binding of Trifloxystrobin in the Cyt b binding pocket supports differential Trifloxystrobin vs Buparvaquone binding 40.

Additional Remarks: no binding had been experimentally demonstrated and yet the word is still used repeatedly and appears not to be by chance. Please revise.

Reviewer #2 (Remarks to the Author):

I feel that the authors have adequately responded to the questions raised by the reviewers and I recommend publication of this article.

Reviewer #3 (Remarks to the Author):

The authors have answered all my points and the manuscript based on my review can be accepted for publication.

Communications Biology COMMSBIO-21-2654C**Trifloxystrobin blocks the growth of *Theileria* parasites and is a promising drug to treat Buparvaquone resistance****Replies to Reviewers' comments:**

We are grateful to the three reviewers for their insightful comments. We list our modifications to the manuscript below.

Reviewer #1 (Remarks to the Author):

Major problem:

I was very surprised to see that my very strong recommendation to move the in silico 3D-hypothetical model of TaCytB (Figure 8) out of the Results section to the Discussion section was not followed. The reason given was that a hypothetical in silico model could be useful for animating discussion, but should not be considered a result based on experimentally derived binding data.

We have addressed this major problem by removing Figure 8c from the Results section. We have added Supplementary Figure 2. As requested by the Reviewer, we comment on the hypothetical model in the Discussion.

Also, I strongly recommended that the discussion should include mention of alternative potential mechanisms that might explain why *T. annulata* is sensitive to Trifloxystrobin and remains sensitive even when TaCytB is mutated and resistant to buparvaquone.

As requested by the Reviewer, we added a comment in the Discussion that MMV668754/Trifloxystrobin could have other unidentified targets.

Moreover, in spite of this low level of conserved amino acid sequence between parasite and chicken CytB, a lack of caution was not evident in the revision that liberally interchanges hypothetical “docking” for “binding” and examples in the text are listed below.

As requested by the Reviewer, the word ‘binding’ now only appears in sentences that have the word ‘predicted’.

Abstract:

Line 40: Furthermore, Trifloxystrobin and Buparvaquone showed distinct binding affinities to parasite Cytochrome B.

As requested by the Reviewer, we have modified the text so that the abstract no longer refers to binding affinities.

Discussion:

Lines 259-260: Furthermore, mutations in the parasite CytB gene associated with Buparvaquone-resistance did not generate resistance to MMV668754/Trifloxystrobin, probably because they bind differently to the q0 site.

As requested by the Reviewer, we have modified the text to clarify that this is based on modeling prediction and not binding studies and we have removed the word 'probably'.

Lines 266 – 269: While our predicted structure is based on published Cytochrome b models from other species (in fact only chicken was used lines 401/402) rather than crystallographic data, the characterized binding of Trifloxystrobin in the Cyt b binding pocket supports differential Trifloxystrobin vs Buparvaquone binding 40.

As requested by the Reviewer, we have removed the offending sentences from the Discussion.

Additional Remarks: no binding had been experimentally demonstrated and yet the word is still used repeatedly and appears not to be by chance. Please revise.

As requested by the Reviewer, the word 'binding' now only appears in sentences that have the word 'predicted'.

Reviewer #2 (Remarks to the Author):

I feel that the authors have adequately responded to the questions raised by the reviewers and I recommend publication of this article.

We thank the Reviewer for their support and their patience.

Reviewer #3 (Remarks to the Author):

The authors have answered all my points and the manuscript based on my review can be accepted for publication.

We thank the Reviewer for their support and their patience.